# Age-dependent changes in circulating Tfh cells influence development of functional malaria antibodies in children

Jo-Anne Chan [1,2,3], Jessica R. Loughland[4,5], Lauren de la Parte[6], Satomi Okano[4], Isaac Ssewanyana[7,8], Mayimuna Nalubega[4,7,9], Felistas Nankya[7], Kenneth Musinguzi[7], John Rek[7], Emmanuel Arinaitwe[7], Peta Tipping[5], Peter Bourke[10], Dean Andrew[4], Nicholas Dooley[4,11], Arya SheelaNair[4], Bruce D. Wines [1,2,12], P. Mark Hogarth[1,2,12], James G. Beeson [1,3,13], Bryan Greenhouse [14], Grant Dorsey[14], Moses Kamya[7], Gunter Hartel [4], Gabriela Minigo [5,15], Margaret Feeney [14], Prasanna Jagannathan [6] & Michelle J. Boyle [1,4,5,9,11✉]

T-follicular helper (Tfh) cells are key drivers of antibodies that protect from malaria. However, little is known regarding the host and parasite factors that influence Tfh and functional antibody development. Here, we use samples from a large cross-sectional study of children residing in an area of high malaria transmission in Uganda to characterize Tfh cells and functional antibodies to multiple parasites stages. We identify a dramatic re-distribution of the Tfh cell compartment with age that is independent of malaria exposure, with Th2-Tfh cells predominating in early childhood, while Th1-Tfh cell gradually increase to adult levels over the first decade of life. Functional antibody acquisition is age-dependent and hierarchical acquired based on parasite stage, with merozoite responses followed by sporozoite and gametocyte antibodies. Antibodies are boosted in children with current infection, and are higher in females. The children with the very highest antibody levels have increased Tfh cell activation and proliferation, consistent with a key role of Tfh cells in antibody development. Together, these data reveal a complex relationship between the circulating Tfh compartment, antibody development and protection from malaria.

[1] Burnet Institute, Melbourne, VIC, Australia. [2] Department of Immunology, Central Clinical School, Monash University, Melbourne, VIC, Australia. [3] Department of Medicine, The University of Melbourne, Parkville, VIC, Australia. [4] QIMR-Berghofer Medical Research Institute, Herston, QLD, Australia. [5] Global and Tropical Health Division, Menzies School of Health Research, Tiwi, Australia. [6] Department of Medicine, Stanford University, Stanford, CA, USA. [7] Infectious Diseases Research Collaboration, Kampala, Uganda. [8] London School of Hygiene and Tropical Medicine, London, UK. [9] Faculty of Medicine, University of Queensland, Brisbane, QLD, Australia. [10] Division of Medicine, Cairns Hospital, Manunda, QLD, Australia. [11] Griffith University, Brisbane, QLD, Australia. [12] Department of Clinical Pathology, The University of Melbourne, Parkville, VIC, Australia. [13] Department of Microbiology, Central Clinical School, Monash University, Melbourne, VIC, Australia. [14] University of California San Francisco, San Francisco, CA, USA. [15] College of Health and Human Sciences, Charles Darwin University, Darwin, NT, Australia. ✉email: michelle.boyle@qimrberghofer.edu.au

*P*lasmodium falciparum malaria remains a leading cause of morbidity and mortality in children, and malaria control programs have been further negatively impacted by the SARS-CoV-19 pandemic[1]. No vaccine yet approaches the efficacy of naturally acquired immunity, which provides nearly complete protection against symptomatic disease in older children and adults living in highly endemic areas. In such settings, the response to infection evolves slowly with age from acute, high-density, symptomatic infections in young children, to infections that are mostly lower density and asymptomatic in older children and adults. Immune development independently increases with both cumulative exposure and host age[2,3]. A better understanding of the mechanisms underpinning immunity and the factors that influence how immunity is acquired is needed to inform the development of improved vaccines and therapeutics that target the immune response.

An important mediator of protective immunity is antibodies that target the parasite to limit replication and parasite burden. Antibody development is driven by CD4 T-follicular helper (Tfh) cells that play key roles in germinal centres to activate naïve B cells and induce memory and antibody-producing B cells[4]. The key role of Tfh cells in driving antibody development has been confirmed in multiple animal studies of *Plasmodium* infection[5–7]. In humans, circulating Tfh cells that resemble germinal centre Tfh has been identified from peripheral blood[8], and can be subdivided into Th1-, Th2- and Th17-like subsets based on CXCR3 and CCR6 chemokine expression[9,10]. Different Tfh subsets have been associated with antibody induction, depending on the infecting pathogen or vaccine, and the context of exposure. During experimental human malaria in previously malaria-naive adults, while all subsets of Tfh cells are activated during infection, Th1-Tfh subsets dominated the response[11], consistent with a Th1-like phenotype of Tfh cell activation in animal models[7,12,13]. Despite the Th1-Tfh cell dominance, in human experimental infection only activated Th2-Tfh cells, and not other Tfh subsets were associated with the induction of anti-malarial antibodies[11]. We have recently shown that activation of Tfh cells during malaria is influenced by age, with adults having greater Tfh activation during infection, particularly within the Th2-Tfh cell subset. In contrast, in children with malaria, activation is restricted to Th1-Tfh cells[14,15]. The impact of this age-dependent activation of Tfh cells on antibody development in malaria is unknown.

While antibodies have been recognised as essential mediators of protection from human malaria for over 50 years[16], more recent studies have shown that the functional capacity of antibodies can be a better correlate of protective immunity than their quantity/titre[17]. Antibodies mediate a variety of effector functions, including fixing complement proteins and interacting with Fcγ receptors (FcγR) to promote opsonic phagocytosis and antibody-dependent cellular cytotoxicity. The importance of these antibody effector mechanisms is established across all parasite life stages. For example, complement-fixing antibodies targeting liver and blood stages have been associated with protection from malaria[18–20], and are a critical mediator of transmission-blocking immunity[21–23]. Opsonic phagocytosis and FcγR-dependent functions have also been associated with protection from the liver and blood-stage infection[24–27]. Little is known regarding the relative kinetics of acquisition of specific functional antibodies across different stages of the parasite as immunity develops.

Due to the central role of Tfh cells in antibody development, targeting Tfh to maximise the induction of functional antibodies may be key to achieving highly efficacious malaria vaccines[17,28]. Thus, it is important to identify factors that influence Tfh cell development, particularly in children with high malaria exposure, and to determine how Tfh development and activation influence

antibody acquisition. Here, we examined Tfh cells and functional antibodies against multiple parasite stages in Ugandan children ages 0–10 with high malaria exposure, and investigated the impact of age and malaria on immune development. Our findings reveal important host and malaria impacts on both Tfh cells and antibody responses that inform vaccine development.

## Results

**Circulating Tfh cell subsets change dramatically with age, independent of malaria.** We analysed Tfh cells (CXCR5+ PD1+) in children enroled in a longitudinal observational cohort in Nagongera, eastern Uganda[29]. We focused our studies on CXCR5+ PD1+ cells as this CD4 T cells subset produces the largest ammounts of Tfh cytokine IL21, and most strongly resembles germinal centre Tfh cells at the transcriptional level[10]. For the present study, samples were collected between January and April 2013 during a period of high malaria burden, with an average entomological inoculation rate of 215 infectious bites per person per year[30]. Tfh cells were quantified by flow cytometry in 212 children aged 0-11 years (54% male) (Supplementary Fig. S1). Consistent with the very high exposure in this cohort, 42% of children had asymptomatic *P. falciparum* infection at the time of sampling. Individual *P. falciparum* infection risk was assessed by household mosquito exposure (mosquito counts performed in individuals' homes, monthly), which correlates with infection in this cohort[2,30,31] (Supplementary Table S1).

Tfh cells (CXCR5+ PD1+) increased as a proportion of all CD4 T cells with age, (Fig. 1A). The proportion of Tfh cells was not impacted by current asymptomatic parasitemia, household mosquito exposure, or sex (Fig. 1B–D). Unexpectedly, within the Tfh cell compartment, there was a dramatic redistribution of Tfh subsets with age, with an increase in Th1-Tfh and a decrease of Th2-Tfh cells between ages 0–7 years (Tfh subsets analysed based on CXCR3 and CCR6 expression. Th1- CXCR3+ CCR6−, Th2–CXCR3−CCR6−, Th17–CXCR3−CCR6+, Fig. 1E). Similar remodeling occurred in the total CXCR5+ (PD1−/+) population and CXCR5− CD4 T cells, however, the proportion of Th1-like cells was significantly higher and Th2-like cells significantly lower in Tfh cells, suggesting that Tfh cell subset redistribution was driven by both global impacts of age on CD4 T cells, as well as Tfh-specific factors (Supplementary Fig. S2). Current infection and sex had no impact on Tfh subset distribution (Fig. 1F/H). There was a suggestion that household mosquito exposure was associated with the proportion of Th17-Tfh cells, however, this was not dose-dependent (reduced Th17-Tfh in children with >40–80 compared to those with >80 household mosquitos/day (Dunn test FDR adjusted $p = 0.04$, Fig. 1G). FoxP3+ Tfh (TfReg cells), have important roles in regulating germinal centre Tfh cells[32]. However, how these responses related to FoxP3+ Tfh cells in the periphery is unclear, and here the proportion of FoxP3+ Tfh cells was not modified by age, current infection, or household mosquito exposure, but there was a trend to higher FoxP3+ Tfh cells in males (Supplementary Fig. S3).

To determine whether the change in Tfh subset distribution within our Ugandan cohort was driven by age or unaccounted for levels of malaria infection (which are co-linear with age in high transmission settings) and/or overall infectious diseases burden, we assessed Tfh cells within healthy, malaria naïve children and adults from low infectious diseases burden community within Australia. Consistent with an age-driven change in the distribution of Tfh subsets, there was a marked decrease in Th2-Tfh cells with age in malaria naïve populations. This decrease was driven by a marked decline in children which then stabilised in adulthood ($R = -0.74$, $p = 0.01$ in children aged 0–15 and $R = -0.19$, $p = 0.6$ in adults age 15 years or greater).

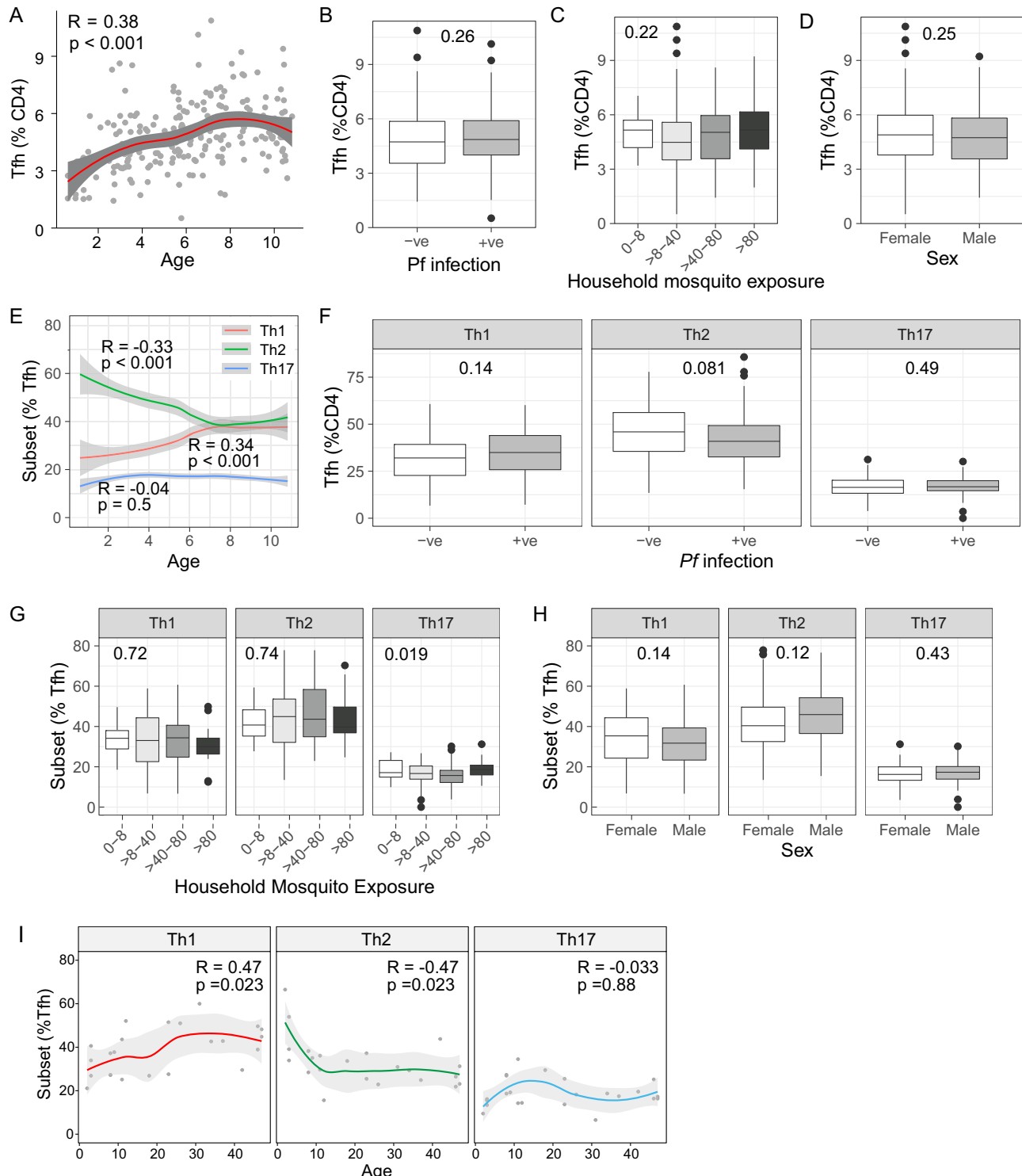

**Fig. 1 Impact of age and infection on Tfh cells subset composition. A–H** Tfh cells were analysed by flow cytometry in $n = 212$ Ugandan children. **A–D** Tfh cells were identified as CXCR5+ PD1+ CD4 T cells. The relationship of the proportion of Tfh cells within the CD4 T cell compartment with age (**A**); current asymptomatic *P. falciparum* infection (**B**); household mosquito exposure (**C**); and sex (**D**). **E–G** Tfh cells were analysed based on CXCR3 and CCR6 expression into Th1 (CXCR3+ CCR−), Th2 (CXCR3−CCR6−) and Th17 (CXCR3−CCR6+) subsets. The relationship of the proportion of Tfh subsets within the Tfh compartment with age (**E**); current asymptomatic *P. falciparum* infection (**F**); household mosquito exposure (**G**); and sex (**H**). **I** Tfh cells subsets were analysed by flow cytometry in malaria naïve children and adults ($n = 23$) from a low infection burden setting in Australia and the relationship with age was assessed. **A**, **E**, and **I** Solid lines are LOESS fit curves with error bands of 95% confidence interval. Spearman rho and *p* are indicated. **B**, **D**, **F** and **H** Mann–Whitney *U* test indicated. **C/G** Kruskal–Wallis indicated. Box and whisker plots, box indicates first and third quartiles for hingers, median line, and whiskers are lowest and highest values no further than 1.5 interquartile range from hinges. Data beyond whisker lines are indicated with points and are treated as outliers. See also Supplementary Figs. S1 and S2. All statistical tests are two-sided, with no adjustment for multiple comparisons. Source data are provided as a Source Data file.

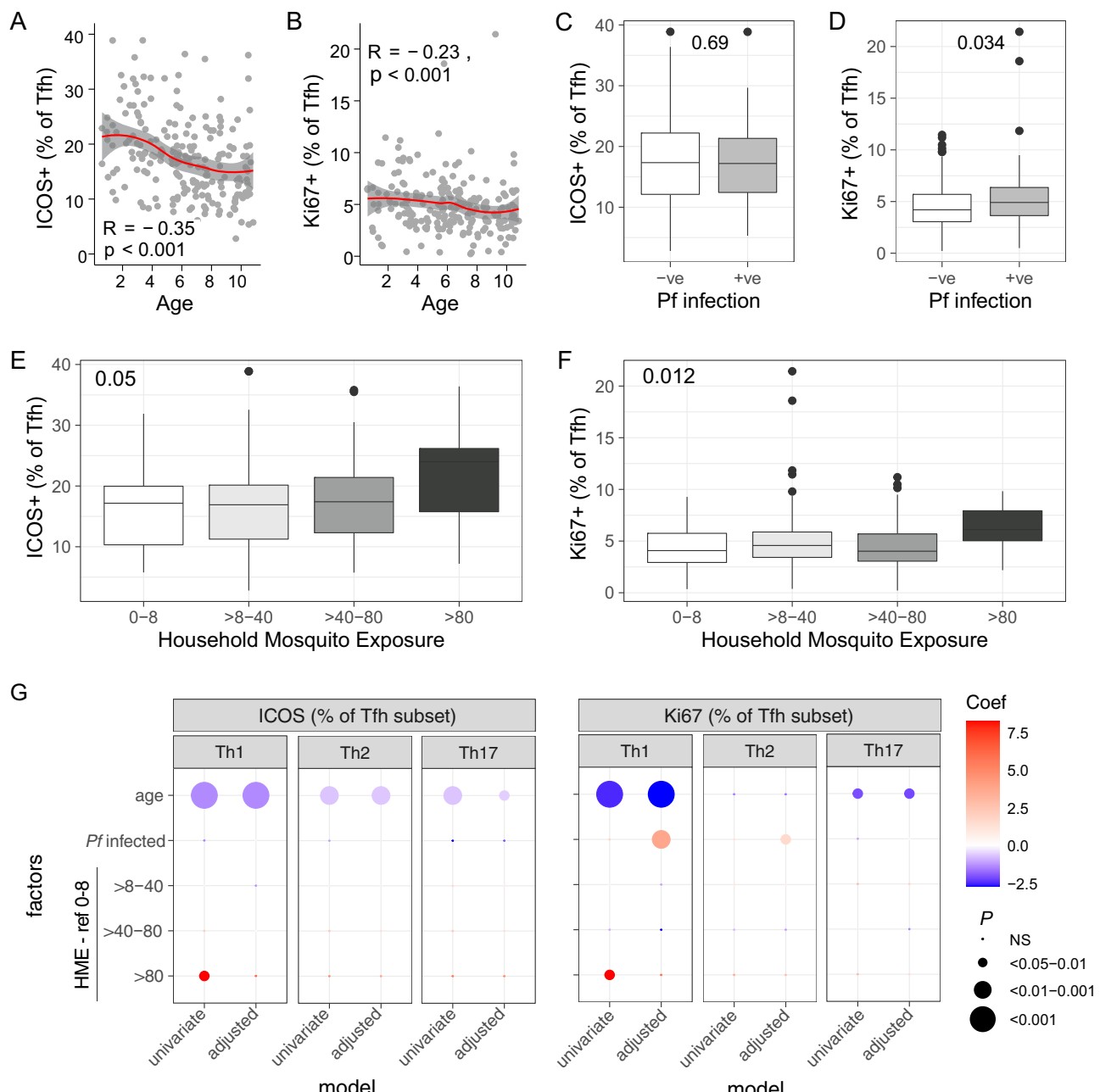

**Fig. 2 Impact of age and malaria on Tfh cell activation and proliferation.** Activation and proliferation of Tfh cells was measured by ICOS and Ki67 expression in 212 Ugandan children. **A**, **B** The relationship between ICOS+ and Ki67+ on Tfh cells with age. Line is LOESS fit curves with error bands of 95% confidence interval. Spearman's rho and *p* indicated. **C**, **D** The relationship between ICOS and Ki67 and current asymptomatic infection. Mann–Whitney *U* test indicated. **E**, **F** The relationship between ICOS and Ki67 on Tfh cells and household mosquito exposure. Kruskal–Wallis indicated. Box and whisker plots, box indicates first and third quartiles for hingers, median line, and whiskers are lowest and highest values no further than 1.5 interquartile range from hinges. Data beyond whisker lines are indicated with points and are treated as outliers. **G** Linear regression model coefficients for the relationship between ICOS and Ki67 on each Tfh subset, and age, Pf infection, and household mosquito exposure (HME). All statistical tests are two-sided, with no adjustment for multiple comparisons. Source data are provided as a Source Data file.

Concurrently, Th1-Tfh cells increased with age in malaria naïve populations (Fig. 1I). Taken together, these data suggest that the Tfh subset distribution undergoes marked changes during childhood, independent of malaria infection.

**Tfh cell activation and proliferation are influenced by both age and *P. falciparum* infection.** During activation, Tfh cells upregulate inducible co-stimulator (ICOS) and the intracellular proliferation marker Ki67, which are required for Tfh cell function

and development[33–35]. As such, we assessed the impacts of age and malaria on ICOS and Ki67 expression on Tfh cells. Both ICOS+ and Ki67+ expression on Tfh cells decreased with age (Fig. 2A, B). The decrease in ICOS expression with age was unique to Tfh cells and not seen on total CD4 T cells, while Ki67 expression also decreased on CD4 T cells (Supplementary Fig. S4A, B). Asymptomatic parasitemia was associated with a significant increase in Ki67 but not ICOS on Tfh cells (Fig. 2C, D). Neither marker was increased in an infection on total CD4 T cells (Supplementary Fig. S4C, D). High household mosquito

exposure (>80 mosquitos/household/night) was associated with increased ICOS and Ki67 on Tfh cells (Dunn test FDR adjusted ICOS > 80 compared to 0–8 $p = 0.05$, compared to >8–40 $p = 0.04$, Ki67 > 80 compared to 0–8 $p = 0.02$; compared to >8–40 $p = 0.02$, compared to >40–80 $p = 0.009$, Fig. 2E, F), but not total CD4 T cells (Supplementary Fig. S4E, F). Sex had no impact on ICOS or Ki67 expression ($p = 0.15$ and $p = 0.72$, respectively).

Because of the interplay of age, current asymptomatic malaria infection, and household mosquito exposure[30], we used linear regression models to determine associations between these factors and ICOS/Ki67 expression in each Tfh subset. Age was associated with declining ICOS expression on Th1, Th2, and Th17 Tfh cells (Fig. 2G, Supplementary Table S2). In contrast, Ki67 expression decreased with age on Th1- and Th17-, but not Th2-Tfh cells. Ki67 but not ICOS expression was higher on both Th1- and Th2-Tfh cells in children with a current asymptomatic infection (Fig. 2G, Supplementary Table S2). The highest household mosquito exposure (<80) was associated with increased ICOS and Ki67 expression on Th1-Tfh cells in univariate analysis (Fig. 2G, Supplementary Table S2). While this association was not maintained once controlling for age and current infection, increasing household mosquito exposure was associated with increased ICOS expression on Th1-Tfh cells when included as a linearised variable, suggesting that higher exposure may contribute to the activation of Th1 cells (Coef 1.87, $p = 0.04$). Sex had no impact on ICOS or Ki67 on any Tfh subset (for ICOS, Th1 $p = 0.18$, Th2 $p = 0.21$, Th17 $p = 0.096$ and for Ki67, Th1 $p = 0.78$, Th2 $p = 0.6$, Th17 $p = 0.78$, respectively). Thus, extending the previous observation that Tfh activation and proliferation during symptomatic malaria in children is restricted to Th1-like Tfh cells[14,15], asymptomatic parasitemia is associated with increases in proliferation in both Th1- and Th2-Tfh cell subsets.

**Hierarchical acquisition of antibodies is dependent on parasite stage and antibody type and function.** To investigate the relationship between Tfh cell compartment changes and antibody development, we comprehensively characterised malaria antibody responses in the cohort described above and an additional 50 children concurrently enroled in the same study (total $n = 262$). We measured antibodies to immunodominant blood stage (MSP2, AMA1), sporozoite stage (CSP) and gametocyte stage (Pfs230) antigens and evaluated the levels of IgG1, IgG3, and IgM, along with functional antibodies that fixed complement, bound Fcγ receptors FcγRIIa and FcγRIII, and had the capacity to mediate opsonic phagocytosis by THP-1 monocytes (OPA). We have recently shown that opsonic phagocytosis by THP-1 monocytes is largely mediated by FcγRI[36]. As expected, the prevalence and magnitude of both IgG and IgM antibodies increased with age (Fig. 3A, B). Antibodies were first acquired to blood-stage antigens MSP2 and AMA1, followed by sporozoite stage CSP, and finally gametocyte stage Pfs230. The large majority of children were seropositive to MSP2 and AMA1. In contrast, the majority of children had little antibody responses to gametocyte antigen Pfs230, with <50% seroprevalence for all antibody types even in older children (Fig. 3A). The low acquisition of gametocyte targeting antibodies is likely due to the very low antigen exposure to gametocytes, with only one child with detectable gametocytes by blood smear at time of sampling. For the blood-stage antigen MSP2, IgG1 was the predominant subclass in young children but switched to predominantly IgG3 with increasing age, consistent with the previous publications[37–41]. In contrast, IgG1 was the predominant subclass for AMA1 across all ages. For both blood-stage antigens MSP2 and AMA1, opsonic

phagocytosis antibodies were acquired first, followed by FcγRIII and then FcγRIIa-binding antibodies. C1q-fixing antibodies were less prevalent (25.8% and 33.3% seropositive for MSP2 and AMA1, respectively). Functional antibodies mediating OPA were also acquired first for CSP. However, antibodies targeting CSP were incapable of binding dimers of FcγRIIa or FcγRIII. There were very low levels of FcγRIIa and FcγRIII-binding, and no C1q-fixing antibodies targeting Pfs230.

To investigate the relationship between antibody responses, we calculated the correlation between magnitudes of antibodies of each type to each antigen (Fig. 3C). For MSP2, IgG3 levels were strongly correlated with C1q, FcRIIa and FcRIII, suggesting IgG3 may be a mediator of these functional responses. In contrast, for AMA1, IgG1 was more tightly correlated with C1q, FcRIIa and FcRIII than IgG3, suggesting that for AMA1 targeted antibodies IgG1 may be the dominant antibody driving functional responses. For CSP targeted antibodies, IgG3 and IgG1, and to a lesser extent IgM all correlated with opsonic phagocytosis levels, suggesting multiple antibody isotypes/subclasses may be involved with mediating phagocytosis of sporozoite stage parasites. In contrast, while IgM has recently been identified as a mediator of opsonic phagocytosis of merozoites[42], MSP2 IgM levels were not correlated with MSP2 opsonic phagocytosis. Additionally, AMA1 IgM was only weakly correlated with opsonic phagocytosis, suggesting that different targets may be involved in IgM-mediated opsonic phagocytosis of merozoites. IgM antibodies were tightly associated across all antigens (MSP2, AMA1, CSP and Pfs230), suggesting that IgM acquisition is con-current across all parasite life cycle stages. There was also a correlation between antibodies that mediate opsonic phagocytosis by monocytes to the blood-stage antigens MSP2 and AMA1. For antibodies targeting Pfs230, there was a strong correlation between IgG3 with opsonic phagocytosis and IgG1 with FcRIII binding, suggesting different relationships between IgG subtypes and functions targeting this gametocyte antigen.

Taken together, these data highlight the complex relationships between antibody isotype/subtype and function, and the differential acquisition of antibodies across parasite stage and functional type. Our data suggest an age-driven hierarchical acquisition of antibodies, which is influenced by both parasite stage (merozoite>sporozoite>-gametocyte) and function/subclass (for MSP2—OPA>IgG3/IgG1>FcRIII>FcRIIa>IgM>C1q; for AMA1—OPA>IgG1>FcRIII>F-cRIIa>IgG3>IgM>C1q; for CSP—OPA>IgG1>IgM>IgG3>C1q; and for Pfs230—IgM>OPA>IgG3>IgG1>FcRIIa>FcRIII).

**Antibody responses are higher in children with current infection and in females.** Previous studies have consistently shown that antibody responses are boosted during infection. Consistent with this, the majority of antibody responses to blood-stage antigens MSP2 and AMA1 were higher in children with a current asymptomatic *P. falciparum* infection (MSP2 IgG3, FcRIII, OPA; AMA1 IgG1, IgG3, C1q, FcRIIa, FcRIII, OPA, Fig. 4A). There was also some boosting to non-blood stage antigens, with increased OPA antibodies to both CSP and Pfs230 during infection. IgM antibodies were not increased to any antigens in currently infected children, in contrast with previous findings by ourselves and others[43,44]. Household mosquito exposure was not associated with antibody responses (Supplementary Fig. S5). Sex influenced multiple antibody responses, with higher IgG3 and C1q to MSP2 and AMA1, and FcRIIa to MSP2 in female children (Fig. 4B).

**Tfh activation and proliferation are associated with functional malaria-specific antibodies.** To investigate whether Tfh phenotypes are associated with malaria-specific antibody development, we performed an unbiased principal components analysis (PCA)

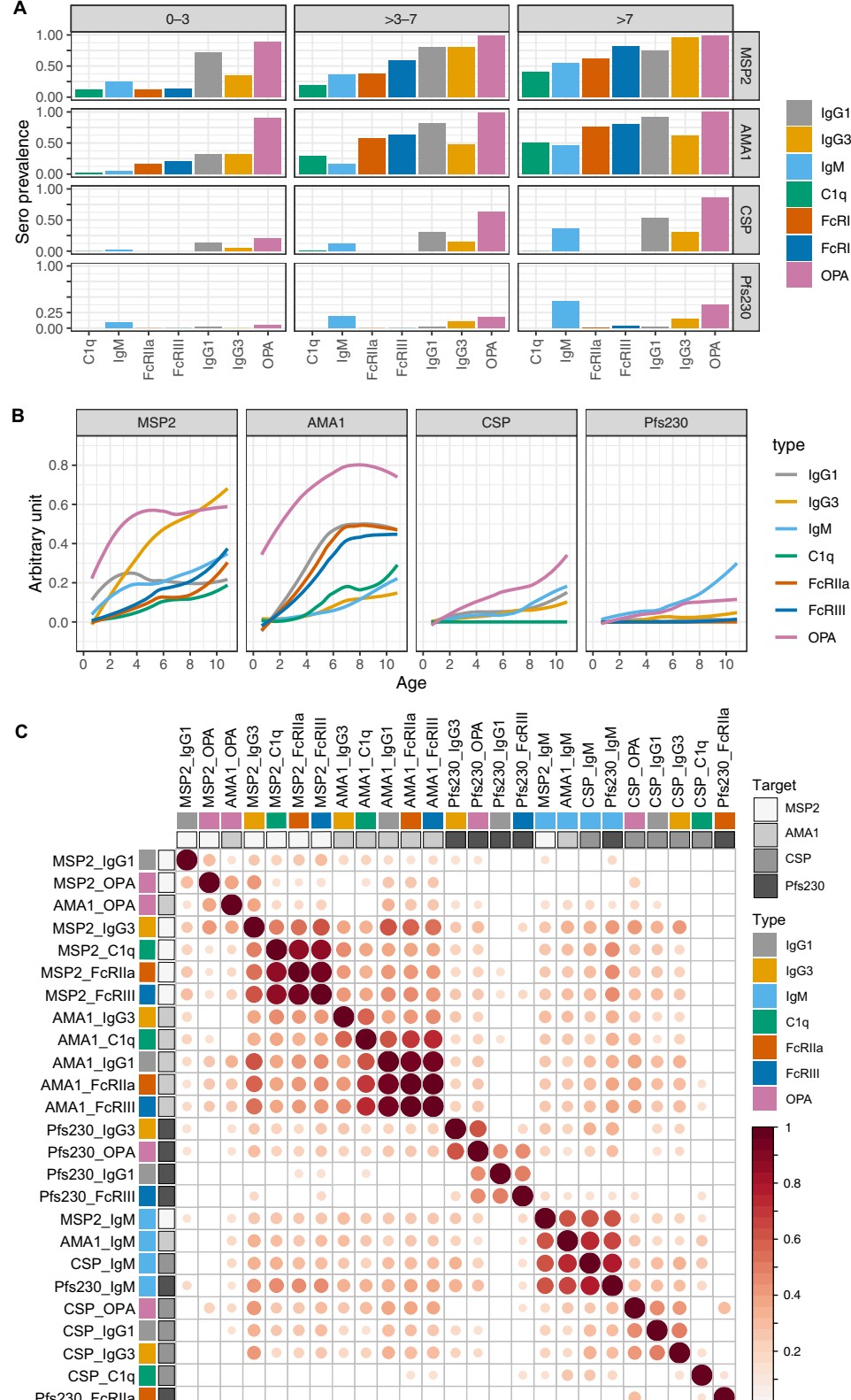

incorporating all Tfh and antibody responses from all individuals with complete data ($n = 211$). This analysis identified four clusters of individuals (Fig. 5A) that differed with respect to age and infection, but not household mosquito exposure and sex (Fig. 5B–E). Children in cluster 1 were significantly younger than in other clusters, followed by cluster 2 children, however, there was no difference between clusters 3 and 4 (Fig. 5B). Current

infection was also associated with cluster (Chi-square test $p = 0.05$), however infection between clusters 3 and 4 was comparable (Fig. 5C). Proportion of Tfh cells (in CD4 T cells), the proportion of Tfh subsets (in Tfh cells), and ICOS and Ki67 expression (in total or Tfh subset) on clusters 1–3 was consistent with age-driven changes, that was associated with increased Tfh cells within CD4 T cells, increased Th1-Tfh and decreased Th2-

**Fig. 3 Seroprevalence and magnitude of antibodies with age. A** Seroprevalence of antibody responses in $n = 262$ children age 0–3, >3–7 and >7 years for IgG1, IgG3, IgM and functional antibodies that mediated complement fixation (C1q), cross-linked FcRIIa and FcRIII and mediated opsonic phagocytosis (OPA) to blood-stage (MSP2, AMA1), sporozoite stage (CSP) and gametocyte (Pfs230) antigens. Seroprevalence is defined as antibody levels greater than mean + 3 SD of negative controls from malaria naïve donors. **B** Magnitude of antibodies to each antigen versus age with line from LOESS fit curves. Antibody magnitude is expressed as arbitrary units, which are calculated by thresholding data at positive seroprevalence levels and scaling data to a fraction of the highest responder. **C** Pairwise Pearson's correlations between the magnitudes of each antibody type/function to each antigen. Blank if correlation coefficient $p > 0.05$. Rows and columns are clustered using hierarchical clustering with Ward.D method. Source data are provided as a Source Data file.

Tfh, and reduced ICOS and Ki67 (Fig. 5F, G). In comparing cluster 3 to cluster 4 (which had comparable age distribution of individuals), cluster 4 had significantly reduced proportions of Th17-Tfh cells, increased ICOS on total Tfh and all subsets, and increased Ki67 on Th2-Tfh cells. The increase of Ki67 on Th2-Tfh cells in cluster 4 was not driven by current parasite infection, with no difference in Ki67 expression on Th2-Tfh cells within cluster 4 between parasite-positive and negative children ($p = 0.26$). Individuals in cluster 4 also had the very highest levels of cytophilic and functional antibodies compared to other clusters, including cluster 3. The Tfh cell differences between clusters 3 and 4 were unique to Tfh cells and not seen in total CXCR5+ or CXCR5− CD4 T cells; there were no differences in subset distribution of these populations between clusters 3 and 4, and while there was a small decrease in Ki67+ CXCR5+ Th17-like cells in cluster 4, no other differences in ICOS or Ki67 between these clusters were observed (Supplementary Fig. S6). Together, PCA and clustering analysis is consistent with a strong impact of age on both Tfh and antibodies. However, this analysis also identifies two distinct clusters of older children, with individuals in cluster 4 with the highest antibodies also having increased ICOS expression on Tfh cells, and Ki67+ Th2-Tfh cells compared to similarly aged children in cluster 3. This finding is consistent with an important role of Th2-Tfh cell proliferation and Tfh activation in malaria antibody development[11,45,46].

**Associations of Tfh subsets and antibodies with protection from malaria.** Finally, to determine the importance of Tfh cell subsets and antibodies in protective immunity to malaria, we measured their association with *P. falciparum* infection and probability of symptoms given infection in the following year. We first considered the odds of any *P. falciparum* infection (including both sub-microscopic and microscopic infections, and both asymptomatic and symptomatic infections). We found that higher proportions of Th17-Tfh cells, but no other Tfh subsets, were associated with reduced odds of any *P. falciparum* infection in the subsequent year of study. This association remained significant after controlling for age, current infection and household mosquito exposure, which is strongly associated with increased odds of infection in this cohort[31] (Fig. 6A, Supplementary Table S3). For humoral responses, antibodies to blood-stage antigens MSP2 and AMA1 were associated with an increased odds of infection in the following year. This is consistent with previous studies showing that antibodies to blood-stage malaria parasites can accurately measure exposure risk[47,48]. Indeed IgG3, C1q, FcRIIa, FcRIII, OPA targeting MSP2 and IgG1, FcRIIa and FcRIII and OPA targeting AMA1 were all associated with increased odds of infection despite adjustment for age, current infection and household mosquito exposure (Fig. 6B, Supplementary Table S4).

As an alternative approach, we modelled associations between both Tfh subsets and antibodies and the risk of infection using least absolute shrinkage and selection operator (LASSO) regression modelling. This model is suited to selecting important components from multicollinear data and indicated that MSP2

OPA and IgG3, and AMA1 IgG1 antibodies, the proportion Th1-17-Tfh cells and current asymptomatic infection were important predictors that are associated with the incidence of any density infection (Fig. 6C).

We next assessed the association of Tfh subsets and antibodies with odds of symptomatic malaria when infected as a measure of protection from disease. Ki67 on total Tfh cells and Th2- and Th17-Tfh subsets (Ki67 % of Tfh, % of Th2-Tfh and %Th17-Tfh) were associated with increased odds of symptoms when infected after adjusting for age and current infection (Fig. 6D, Supplementary Table S5). For antibody responses, MSP2 (IgG1, IgG3, IgM, OPA), AMA1 (IgG1, IgG3, IgM, FcRIIa, FcRIII, OPA), CSP (IgG1, IgG3, OPA) and Pfs230 (IgG3, IgM, OPA) were all associated with reduced odds of symptoms given infection in the year following. After adjusting for age and current infection, only MSP2 (IgG3) responses remained strongly associated with reduced odds of symptoms when infected ($p = 0.02$, Fig. 6E, Supplementary Table S6). Using LASSO modelling, MSP2 IgG3, AMA1 IgM and CSP OPA antibodies, along with age and asymptomatic infection were selected as factors that best predicted and were associated with reduced incidence of symptoms given infection. In contrast, Ki67 expression on Th17-Tfh and Th2-Tfh cells, and the proportion of Th2-Tfh cells were associated with increased incidence of symptoms given infection (Fig. 6F). Together, these data reveal a complex relationship between the circulating Tfh compartment, antibody development, and protection from malaria. Our findings suggest an important role of cytophilic antibodies to blood-stage antigens in protection from symptomatic malaria.

## Discussion

Antibody induction during infection, including malaria, is driven by T-follicular helper CD4 T cells[49]. However, little is known regarding the factors that influence Tfh cell development in children living in malaria-endemic settings, or their impact on the relative kinetics of acquisition of functional antibodies against different malaria parasite stages. Here, we show that the Tfh cell compartment changes dramatically with age, with a marked decrease in Th2-Tfh cells in the first 10 years of life in both malaria-exposed and malaria-naïve children. In malaria-exposed children, Tfh cell activation and proliferation also increased with age and were further influenced by current *P. falciparum*. In addition, we demonstrate that antibody acquisition with age followed a hierarchical order and that age-dependent changes in antibody subclasses and functions differed with parasite stage and antigen target. Clustering analyses revealed that children with the highest levels of cytophilic antibodies had increased activation of Tfh cells and proliferating Th2-Tfh cells compared to similarly aged children. Importantly, cytophilic and functional antibodies to blood and sporozoite stage parasite proteins were associated with protection from symptoms among those with *P. falciparum* infection. Together, these results have implications for our understanding of Tfh cell biology in humans and antibody-mediated immunity from malaria.

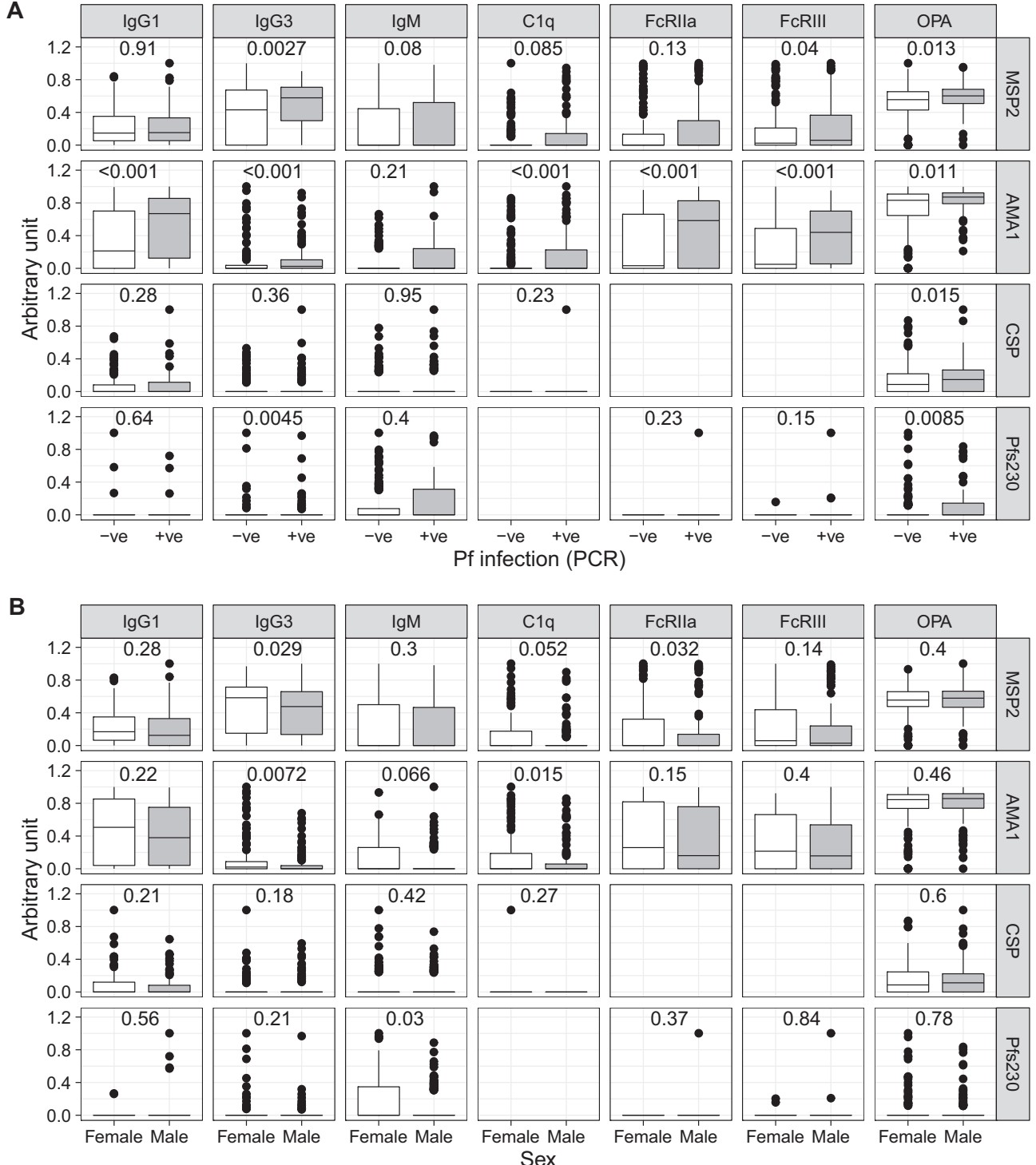

**Fig. 4 Relationships between antimalarial antibodies and current infection and sex.** Magnitude of antibody responses IgG1, IgG3, IgM, and functional antibodies that mediated complement fixation (C1q), cross-linked FcRIIa and FcRIII, and mediated opsonic phagocytosis (OPA) to blood-stage (MSP2, AMA1), sporozoite stage (CSP) and gametocyte (Pfs230) antigens in $n = 262$ children. Antibody magnitude is expressed as arbitrary units, which are calculated by thresholding data at positive seroprevalence levels and scaling data to the highest responder. **A** Magnitude of antibodies in children with and without a current *P. falciparum* infection detected by PCR. **B** Magnitude of antibodies in female and male children. Mann–Whitney *U* tests indicated. Box and whisker plots, box indicates first and third quartiles for hingers, median line, and whiskers are lowest and highest values no further than 1.5 interquartile range from hinges. Data beyond whisker lines are indicated with points and are treated as outliers. Source data are provided as a Source Data file.

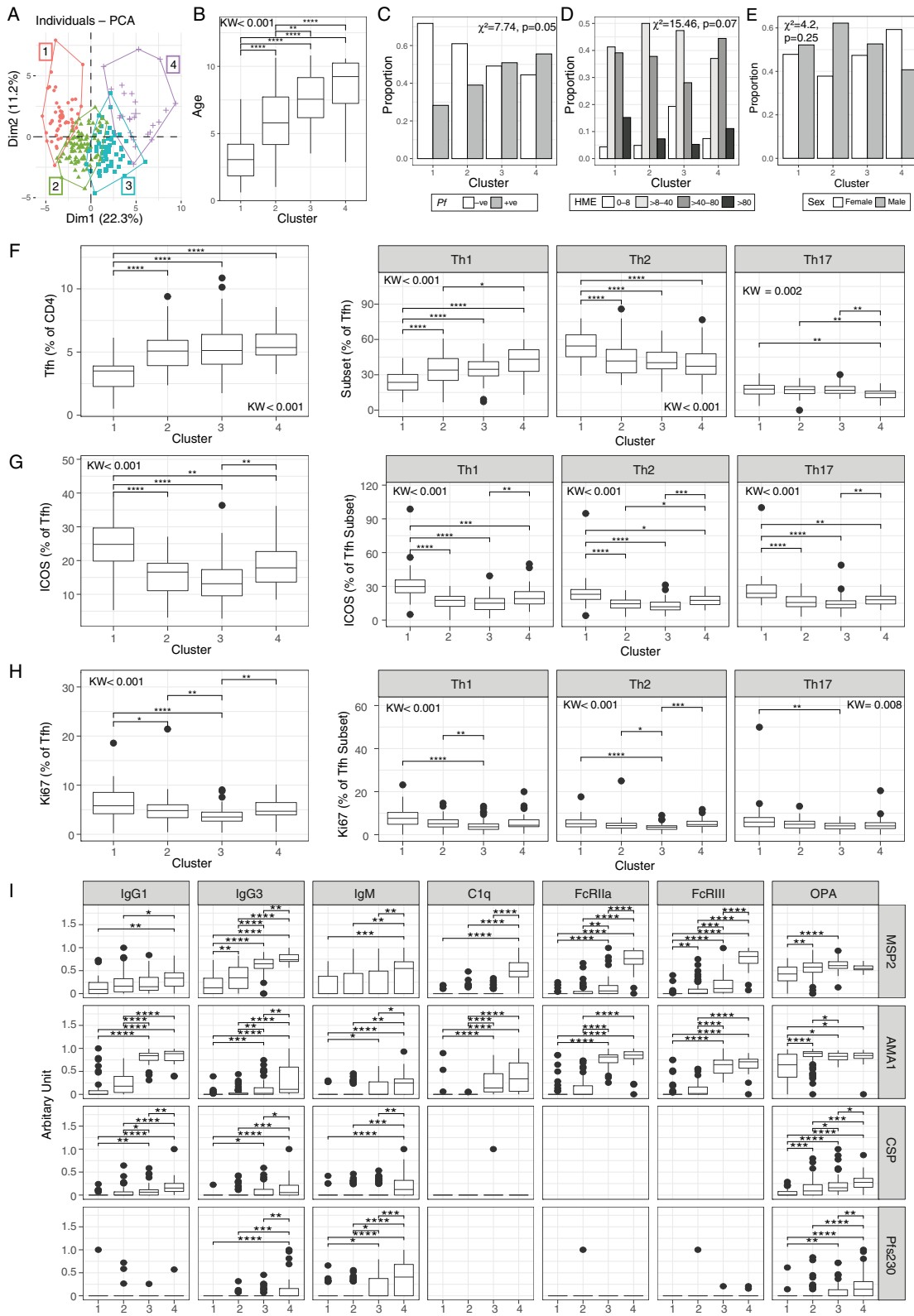

**Fig. 5 Principle component analysis and clustering of individuals. A** Tfh and antibody responses were analysed by PCA and individuals clustered by *k*-means in *n* = 212 children. **B** Age, **C** infection, **D** household mosquito exposure (HME) and **E** sex in *k*-means clusters. **F** Tfh cells and subsets in *k*-means clusters. **G** ICOS and **H** Ki67 on total Tfh and Tfh cell subsets in *k*-mean clusters. **I** Antibody levels for each isotye/subclass and function in each *k*-mean cluster. **B**, **F–I** Kruskal–Wallis and Dunn post-analysis FDR-adjusted indicated. *p* < 0.05, **p* < 0.01, ***p* < 0.001, ****p* < 0.0001. **C**, **D** Chi-square test indicated. Box and whisker plots, the box indicates first and third quartiles for hingers, median line, and whiskers are lowest and highest values no further than 1.5 interquartile range from hinges. Data beyond whisker lines are indicated with points and are treated as outliers. Source data are provided as a Source Data file.

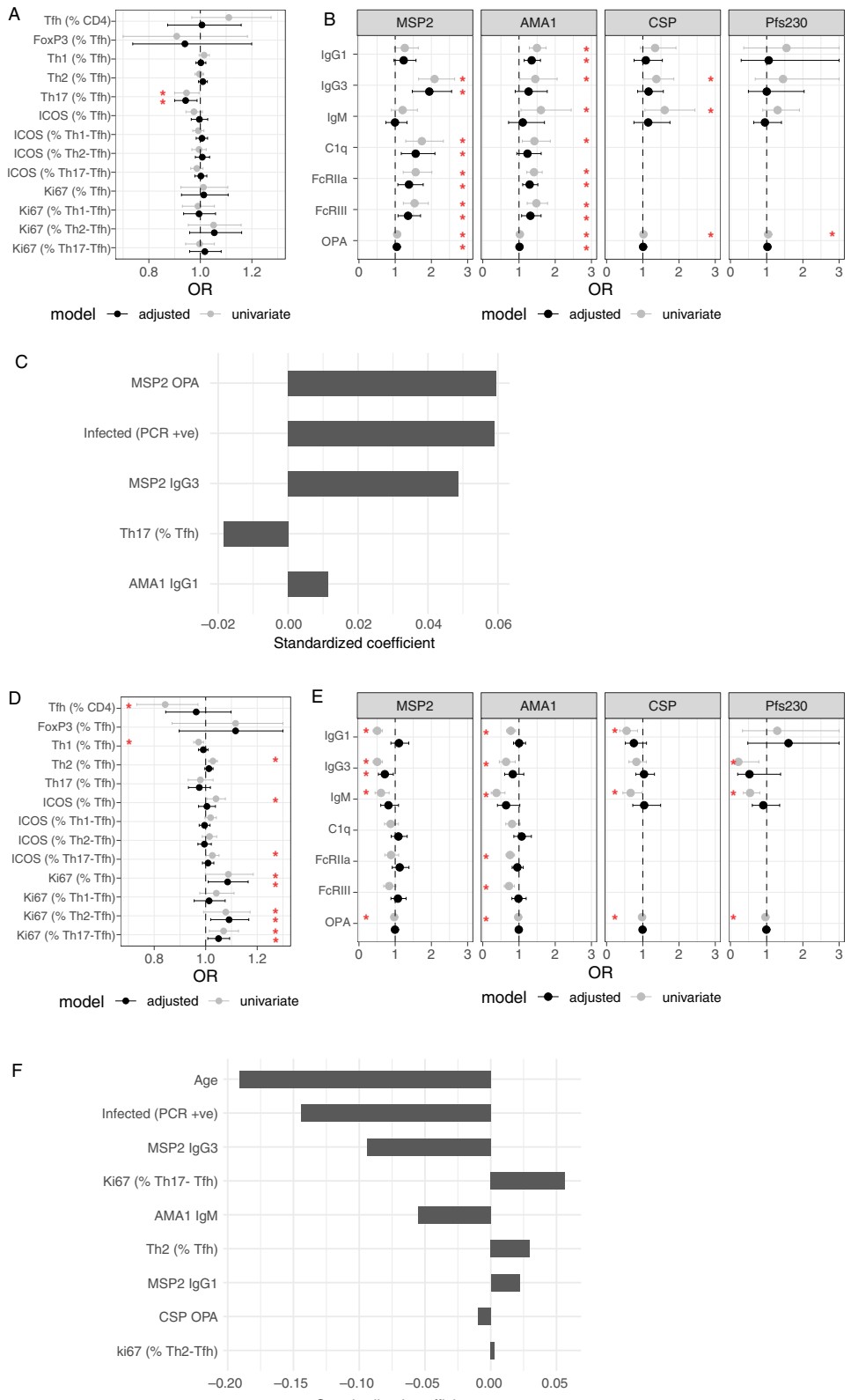

We found that in malaria-exposed and naïve populations, the Tfh cell compartment undergoes marked changes with age, with a significant decline in Th2-Tfh and an increase in Th1-Tfh cells in children. To the best of our knowledge, this is the first report of this age-dependent re-distribution of Tfh subsets in children. The mechanisms driving this dramatic age redistribution of Tfh subsets are unknown but may be connected to the overall maturation of the immune response with age. Indeed, Th1 supporting cytokines and Th1 CD4 T cell responses to infection only reach levels of adult maturity around age 5–10[50]. While the specific mechanisms that mediate Th2-Tfh compared to Th1-Tfh development are largely unknown[51], this maturation of the immune system to support Th1 response may be important. How these changes to Tfh subsets impact immune

**Fig. 6 Association between Tfh cells, antibodies and malaria.** Odds ratios for **A** Tfh subsets and **B** antibodies with any infection (LAMP positive or blood smear positive) in the year following. **C** LASSO Poisson regression standardised coefficients for responses selected as informative prognostic factors for incidence of any infection (LAMP positive or blood smear positive). Odds ratios for **D** Tfh subsets and **E** antibodies with symptomatic infection (blood smear positive with fever, given infection (LAMP positive or blood smear positive) in the year following. **F** LASSO Poisson regression standardised coefficients for responses selected as informative prognostic factors for symptoms given infection. **A**, **B**, **D** and **E** Data are odds ratios and error bars are 95% confidence intervals. Unadjusted data and data adjusted for potential confounders are presented. Red star indicates $p < 0.05$; in **A** and **B**, data were adjusted for age (continuous), current infection detected by PCR and household mosquito exposure (HME), while in **D** and **E**, data were only adjusted for age (continuous) and current infection. Data from A/C is $n = 211$ children, data from **B** is $n = 261$ children, data from **D** and **F** is $n = 207$ children, data from **E** is $n = 255$ children. See also Supplementary Tables S2–5. Source data are provided as a Source Data file.

development with age requires further investigation. Changes to Tfh subset distributions were not further impacted by current asymptomatic infection in our cohort. However, both Th1-Tfh and Th2-Tfh cells had increased proliferation in children with a current asymptomatic infection. In contrast, studies in Mali and Indonesia have shown that only Th1-Tfh cell subsets are activated among children with symptomatic malaria[14,15]. These differences may be due to underlying levels of inflammation between asymptomatic and symptomatic parasite infection, where IFNγ and TNFα are associated with skewing of Tfh to Th1-like subsets during *Plasmodium* infection[12,52]. In addition, multiple antibody responses were also boosted in asymptomatic parasitemia, consistent with Th2-Tfh cell activation and proliferation playing a role in antibody development[11]. Further, clustering analysis revealed that children with the highest antibody levels had relatively higher activation of all Tfh subsets and proliferation of Th2-Tfh cells, compared to similarly aged children. This finding is consistent with an important role of Tfh cells, particularly Th2-Tfh cells, in antibody development in malaria[11].

Understanding the acquisition of antibodies that protect against symptomatic malaria is crucial to informing the development of efficacious vaccines. Our study, for the first time, measured the magnitude and function of antibodies against immunodominant antigens from different parasite life cycle stages (blood, sporozoites and gametocyte antigens). We found that antibodies in this cohort were first acquired to blood-stage antigens such as MSP2 and AMA1, while antibodies to sporozoite CSP and gametocyte Pfs230 were acquired later. These findings are consistent with previous results in which we have shown functional antibodies that bind FcγRIIa and FcγRIII are significantly higher to merozoites compared to CSP antigen in Kenya people[36]. Differences in antibody acquisition across parasite stages are likely driven by antigen burden. Blood stages have the highest parasite burden due to ongoing replication of the parasite within red blood cells, and both MSP2 and AMA1 are abundantly expressed on the surface of merozoites[53]. In contrast, sporozoites are hidden from the immune system due to their gliding motility through hepatocytes. Similarly, gametocytes reside within red blood cells and emerge only within the mosquito midgut, thus reducing their capacity to stimulate an immune response. The extremely slow and infrequent acquisition of functional antibodies to sporozoite stages is consistent with the minimal levels of sterile protection against hepatocyte infection in naturally exposed populations[54]. Similarly, low levels of functional antibodies targeting gametocyte stages suggest that transmission-blocking immunity is low in children and that children could remain a large reservoir of transmission despite their gradual development of immune protection from disease[55]. Alongside the parasite stage, it is clear that the fine specificities at the subsets and functional levels are also influenced by specific characteristics of the antigen. Here, MSP2 IgG3 responses dominated and correlated with the functional capacity of antibodies, in contrast to AMA1 where IgG1 antibodies are dominant. This finding is consistent with previous reports of IgG3

dominance in some but not all merozoite antigens and specific protein epitopes[41,56–58]. While the mechanisms mediating IgG3 or IgG1 dominance to specific parasite proteins are unknown, the presence of specific T-cell epitopes that induce strong IL10 responses alongside IFNγ has been implicated in experimental models[59]. Consistent with an important role of IL10 in skewing antibodies to IgG3, there is an age-associated switch from IgG1 to IgG3 dominance for MSP2 antibodies seen here and in other studies[38,39,41], that coincides with the development of IL10/IFNγ co-producing malaria-specific T cells within these same Ugandan children[31].

In our study, multiple cytophilic and functional antibodies to blood and sporozoite stage antigens were associated with protection from symptomatic infection, consistent with previous studies[19,20,26–28,60]. Sex was also associated with antibody levels, with increased antibodies in female children. While sex-based differences in malaria antibodies have been understudied, this finding is consistent with a recent study in a different cohort of Ugandan children aged 7–16 where female children/adolescents had higher MSP1 and MSP2 antibodies compared to males[61]. Indeed, in other infection and vaccination settings, females generally have greater antibody responses and global IgG levels and B cells[62,63]. A recent study showed that female children have faster rates of parasite clearance compared to males[64]. Hence, sex-based differences in antibody responses may play an important role in protection from malaria. Levels of blood-stage antibodies were also associated with increased odds of infection, this is consistent with previous studies showing that antibodies to blood-stage malaria parasites can accurately measure exposure risk[47,48]. Interestingly, we also observed that increased proportions of Th17-Tfh cells were associated with reduced odds of infection, but not symptoms once infected. While it is possible that Th17-Tfh cells may have a role in protecting from blood-stage infection measured here, we hypothesis that this association is instead driven by exposure. Children with higher malaria exposure have a proportional decrease in Th17-Tfh cells as malaria-specific Th1- and Th2-Tfh subsets expand, consistent with the proliferation of Th1- and Th2- subsets during active infection seen here and reported previously[14,15], and expansion of Th1-Tfh cells in *P. vivax* malaria[65].

There were several important limitations to our findings. First, samples were obtained from Ugandan children at a single cross-sectional timepoint. Although this study represents one of the largest studies to date, future studies will need to perform repeated assessments within children over time to determine the impact of age and malaria exposure on both antibody responses and T cell maturation within individuals. Additionally, the cohort investigated here has very high malaria exposure[29] and further studies are needed in areas of low and moderate transmission to understand these relationships in other populations. Second, the analysis of circulating Tfh responses evaluated bulk and not antigen-specific Tfh cell responses[14]. Antigen-specific cells could not be identified in our studies based on the expression of activation or proliferation markers alone, as these markers were

detected on Tfh cells in both infected and uninfected children, and strongly associated with age. As such, other means of identifying parasite-specific Tfh cells, such as Activation Induced Marker assays[14,66], will be required. Additionally, we were not able to differentiate between PD1[hi] and PD1[intermediate] populations due to the limited sensitivity of flow cytometry reagents. These two populations may have important functional differences in driving antibody development. Although we identified important associations between Tfh populations, age, antibody acquisition and protection, future studies are planned to assess the antigen-specific and functional capacity of Tfh cell response in this population. Additionally, absolute cell counts of CD4 T cells are not available for our cohort. While it is recognised that the concentration of CD4 T cells reduces with age in children, it is unknown whether these changes to cell concentrations are linked to function and/or protection from disease.

In conclusion, in a large cohort of children living in a highly malaria-endemic setting, we describe dramatic redistribution of the Tfh cell compartment from Th2-Tfh to Th1 dominance with age, and associations between Tfh activation, proliferation and concurrent *P. falciparum* infection. Additionally, we have identified the hierarchical acquisition of functional antibodies driven by parasite stage and antibody type/subclass and function, with cytophilic and functional antibodies significantly associated with protection from the symptoms of *P. falciparum* infection. Young children are the most at risk of malaria, and are thus the target of vaccine interventions. As such, these findings are of relevance for our understanding of the development of immunity in this population, and vaccine development for children in endemic areas.

## Methods

### Study population

*Ugandan children cohort.* Samples were obtained from 262 children (117 female/ 145 male) enroled in a longitudinal study by the East African International Centres of Excellence in Malaria Research conducted in the high transmission areas of Uganda[29,31,67]. This cohort consists of 100 households within the rural Nagongera sub-county in the Tororo district, where malaria transmission is holoendemic with seasonal peaks from October to January and April to July. All households within the subcounties were enumerated and mapped using handheld global positioning systems (Garmin e-Trex 10 GPS unit, Garmin International Inc., Olathe, KS). A household was defined as any single permanent or semipermanent dwelling acting as the primary residence for a person or group of people that generally cook and eat together. Using a computerised number generator, random samples of households from each subcounty were approached consecutively, and 100 households were enroled per site into both the entomologic surveys and cohort studies if they met the following criteria: (1) at least one household resident 0.5–10 years of age and (2) at least one adult resident available for providing informed consent. As such, no-self selection bias is involved in this study, and other bias in recruitment is limited. Following the selection of the household, one adult caregiver (>20 years) and all children (eligibility of those aged 6 months to 10 years) were enroled in the study. Upon enrolment, all study participants were given an insecticide-treated bed net and followed for all medical care at a dedicated study clinic. Children who presented with a fever (tympanic temperature >38.0 °C) or a history of fever in the previous 24 hours had blood obtained by finger prick for a thick smear. If the thick smear was positive for *Plasmodium* parasites, the patient was diagnosed with malaria regardless of parasite density and treated with artemether-lumefantrine. Routine assessments were performed in the study clinic every 3 months, including blood smears (which were assessed for both blood-stage and gametocyte parasites) and dry blood spots to detect parasite infection by PCR. Negative blood smears obtained at routine assessments were tested for the presence of submicroscopic malaria parasites using loop-mediated isothermal amplification (LAMP). Blood was drawn from each participant at a single cross-sectional timepoint between January and April 2013. Participant demographics for the current study are in Supplementary Table S1. Participants were reimbursed travel costs for all visits to the study clinic.

*Malaria naïve cohort.* PBMCs were collected from a healthy malaria-naive cohort of children (n = 13, median age 8 IQR [3-13], 38% female) and adults (n = 14, median age 39.5 IQR [25-43], 43% female) from a clinic of hospital outpatients. Volunteers were assessed by an on-site immunologist, where they were confirmed immunologically healthy and malaria-naïve. No compensation was provided for participants in this cohort.

### Ethics statement

Ethics approval for the use of human samples was obtained from the Makeree Univestiy School of Medicine Research and Ethics Committee (2011-167), the Uganda National Council of Science and Technology (HS1019), the University of California, San Francisco Committee of Human Research (11-05995), and the Alfred Health Ethics Committee (#328/17), QIMR-Berghofer Human Research Ethics Committee (P3444 and P3445), Stanford University Institutional Review Board (IRB 41197) and Menzies School of Health Research Human Ethics Research Committees (2012-1766). Written informed consent was obtained from all adult study participants and parents or legal guardians of the children.

### Measuring antibodies to recombinant *P. falciparum* antigens by ELISA

Recombinant MSP2 FC27 was expressed in *E. coli*[68]. PfAMA1[43], full-length PfCSP[69] and Pfs230D1M[70] were previously generated via expression in the mammalian HEK293 cell expression system.

The level of antibodies to recombinant *P. falciparum* antigens and merozoites was measured by ELISA[68]. Antigens were coated at 1 µg/ml in PBS and incubated overnight at 4 °C. Merozoites were coated at $1 \times 10^7$ cells per ml and incubated for 2 h at 37 °C. Plates were blocked with 1% casein in PBS (Sigma-Aldrich) for recombinant antigens and with 10% skim milk in PBS for merozoites, for 2 h at 37 °C. Following that, human antibodies (tested in duplicate) were added. For IgG detection, plates were incubated with a goat anti-human IgG HRP-conjugate (1/ 1000; Thermo Fisher Scientific Cat#2-8420)). For IgG subclasses and IgM detection, plates were incubated with an additional step of mouse anti-human IgG1 (clone HP6069), IgG2 (clone HP6002), IgG3 (clone HP6050), IgG4 (clone HP6025) or IgM (clone HP6083) (1/1000; Thermo Fisher Scientific cat# A-10630, 05-3500, 05-3600, A10651, 054900 respectively) for 1 h at room temperature, followed by detection with a goat anti-mouse IgG HRP (1/1000; Millipore cat #AP308P) for 1 h at room temperature. Colour detection was developed using TMB liquid substrate (Sigma-Aldrich), which was subsequently stopped using 1 M sulfuric acid. Antibodies were diluted with 0.1% casein in PBS for recombinant antigens and with 5% skim milk for merozoites. Serum dilution used for MSP2 FC27 was 1/250 for IgG subclasses and IgM, 1/100 for C1q and FcγR. Serum dilution used for PfAMA1 was 1/250 for IgG subclasses and IgM, 1/100 for C1q and 1/125 for FcγR. Serum dilution used for PfCSP was 1/100 for IgG subclasses, 1/250 for IgM and 1/ 100 for C1q. Serum dilution used for Pfs230D1M was 1/100 for all isotypes/ subclasses and functions. PBS was used as a negative control and plates were washed thrice (with PBS with 0.05% Tween for recombinant antigens and with PBS only for merozoites) in between antibody incubation steps, using an automated plate washer (ELx405, BioTeck, USA). The level of antibody binding was measured as optical density at 450 nm (for TMB) using the Multiskan Go plate reader (Thermo Fisher Scientific).

### Complement fixation assay

The capacity of human antibodies to fix complement C1q was measured using previously optimised methods[19,20]. Antigen coating and blocking were performed as above for standard ELISA. After incubation with human antibody samples, purified human C1q (10 µg/ml; Millipore) was added as a source of complement for 30 min at room temperature, followed by a rabbit anti-C1q (1/2000; in-house, previously generated and validated[69]) detection antibody and finally, a goat anti-rabbit IgG HRP (1/2500; Millipore cat# AB97051) for 1 h at room temperature. The level of C1q-fixation was developed using TMB liquid substrate (Sigma-Aldrich), reactivity was stopped using 1 M sulfuric acid and measured as optical density at 450 nm.

### Measuring antibodies that bind Fc receptors

Measuring antibodies that have the capacity to bind Fc receptors was performed with previously optimised methods[27,36,71]. Antigen coating and blocking were performed as above for standard ELISA. After incubation with human antibody samples, 50 µl of biotinylated recombinant soluble dimers (0.2 µg/ml of dimeric FcγRIIa, 0.1 µg/ml of dimeric FcγRIII; expressed in-house using the HEK293 system) were added to the plates and incubated for 1 h at 37 °C. Subsequently, a streptavidin HRP-conjugated antibody (1/10,000; Thermo Fisher Scientific) was added for 1 h at 37 °C. Colour detection was developed using TMB liquid substrate and measured as optical density at 450 nm.

### Opsonic phagocytosis of antigen-coated beads (OPA)

Measuring opsonic phagocytosis of antigen-coated beads was performed with previously optimised methods[36]. Antigen-coated beads ($5 \times 10^7$ beads/ml) were incubated with serum samples for 1 h at room temperature before washing and co-incubation with THP-1 monocytes for 20 min at 37 °C (THP-1 cells obtained from ATCC, not independently validated, but Fc-gamma-receptor expression was confirmed by Flow Cytometry). The proportion of THP-1 cells containing fluorescent-positive beads was acquired by flow cytometry (FACS Verse, BD Biosciences) and analysed using FlowJo software (version 10). Negative controls from non-exposed Melbourne residents were included in all assays.

### Flow cytometry

Ex vivo Tfh phenotype and activation were assessed by flow cytometry. PBMCs were thawed in 10% FBS/RPMI, and rested for 2 h at 37 °C, 5% $CO_2$. In brief, 1 M PBMCs were stained with surface antibodies to identify Tfh subsets and activation/proliferation. PBMCs were stained for 15 min at RT, washed

with 2% FBS/PBS, for intracellular markers, PBMCs were permeabilised with CytoFix/CytoPerm (BD) and 1X Perm/Wash (BD) and stained with intracellular markers. For Ugandan children samples the following antibodies were used: CD4 (RPA-T4, PerCP Cy5.5, Biolegend 300530, 1/50 dilution), CXCR5 (J252D4, BV711, Biolegend 356934, 1/83.3 dilution), PD-1 (EH12.1, PE, BD Biosciences 560795, 1/16.7 dilution), CXCR3 (1C6 BV421 BD Biosciences 562558, 1/31.25 dilution), CCR6, (11A9 APC-R700, BD Biosciences 565173 1/50 dilution), ICOS (C398.4A, APC-Cy7, Biolegend, 313530, 1/62.5 dilution), Ki67 (B56, FITC, BD Biosciences 556026 1/62.5), FoxP3 (150D, AF647, Biolegend 320014, 1/83.33). For Australian malaria naïve samples the following antibodies were used with 1 million cells stained in 50µl: CD3 (SK7, FITC, Biolegend 344804, 1/10 dilution), CD4 (OKT4, PerCP/Cy5.5 Biolegend 317428, 1/250 dilution), CXCR5 (J25D4, BV711, Biolegend 356934, 1/50 dilution), PD1 (EH12.1 PE-Cy7 BD Biosciences 561272, 1/100 dilution), CXCR3 (IC6, BV421 BD Biosciences 562558, 1/50 dilution), CCR6 (11A9, BV50, BD Biosciences 563922, 1/100 dilution) (Supplementary Table S7). Samples were acquired on Aurora Cytek 3 laser instrument (Australian samples) or an Attune NXT Flow cytometer (Ugandan samples).

***P. falciparum* culture**. *P. falciparum* 3D7 isolates were maintained in continuous culture in RPMI-HEPES medium supplemented with hypoxanthine(370 mM), gentamicin (30 mg/ml), 25 mM sodium bicarbonate and 0.25% AlbuMAX II (GIBCO) or 5% heat-inactivated human sera in O + RBCs from malaria-naive donors (Australian Red Cross blood bank). Cultures were incubated at 37 °C in 1% $O_2$, 5% $CO_2$, 94% $N_2$. Trophozoite and schizont stage parasites were purified by MACS separation (Miltenyl Biotec). Merozoites were isolated from matured schizonts via filtration with previously optimized methods[19].

**Statistical analyses**. All statistical data analyses were performed using Prism 7 (GraphPad), STATA (version 15) and RStudio (R version 4.0.4). All statistical tests are two-sided. For analysis of antibody levels, antibody magnitude is expressed as arbitrary units, which are calculated by thresholding data at positive seroprevalence levels and scaling data to the highest responder. Correlations between antibody variables and age were assessed using Pearson correlation for continuous variables. Clustering of correlations was performed with H-clustering with Ward.D method with R Hmisc package (version 4.7-0). To assess the association between Tfh cells and antibodies once controlling for age, the lower limit of antibodies magnitudes was set at 0.001 and then log-transformed before analysis in linear regression modelling. For clustering analysis of individuals, only those with complete Tfh and antibody data were included. All data were transformed into z-scores and then PCA was performed in factoextra (version 1.0.7) and FactoMineR (version 2.4). Kmeans cluster number chosen in 'useful' package (version 1.0.7) and *k*-means clusters identified with 'stats' (version 4.1.2) package in R. Four clusters were chosen based on Hartigan's rule with an inclusion value of ~10, and sample size consideration.

To assess the odds of infection and disease in the year following with antibody and Tfh subsets, routine assessments with active case detection were performed in the study clinic every 3 months, including blood smears and dry blood spots to detect parasites infections. Negative blood smears obtained at routine assessments were tested for the presence of sub-microscopic malaria parasites using loop-mediated isothermal amplification (LAMP)[72]. At the time of routine assessments, children were classified into four categories as described previously[31,67]: (1) no evidence of parasite infection; (2) asymptomatic, submicroscopic (LAMP positive) infection; (3) asymptomatic, blood smear positive infection, or (4) symptomatic malaria defined as fever with a smear-positive infection, with a window of 21 days prior to and 7 days following the routine visit to ensure capture of malaria episodes that were recently treated or infections that soon became symptomatic. To assess relationships between cross-sectional antibody responses and the prospective odds of infection measured at repeated, quarterly routine assessments, the odds of any infection (defined as either LAMP positive or blood smear positive with or without symptoms) were calculated using multilevel mixed-effects logistic regression, accounting for repeated measures within individuals and clustered on household to account for multiple children in each individual household. In multivariate analysis, odds ratios for infection risk were adjusted for age (linear), current asymptomatic infection and household mosquito exposure (categorical 0–8, >8–40, >40–80, >80 mosquitos/household/day). Household mosquito exposure was calculated as described previously (previously named daily mosquito exposure rate)[31]. For each individual, household mosquito exposure was calculated based on the mean household-level female Anopheles mosquito counts obtained from CDC light traps placed overnight (once per month) within the household of all trial participants[30]. Mean female Anopheles mosquito counts from the prospective 12 months were calculated and used in the analysis as a categorical variable (0–8, >8–40, >40–80, and >80 mosquitos/household/day), based on analyses of the relationship between household mosquito exposure and *P. falciparum* infection where each category is associated with increased odds of infection[31]. Similarly, to assess the odds of symptoms given infection (defined as cases of clinical symptoms when an infection was detected either by LAMP or blood smear) odds were calculated using multilevel mixed-effects logistic regression, accounting for repeated measures within individuals and clustered on household to account for multiple children in each individual household. In multivariate analyses, odds ratios were adjusted for age (continuous) and current infection.

LASSO Poisson regression modelling was used to select the most informative prognostic factors for (1) incidence of density infections and (2) incidence of symptoms when infected across the multiple visits during the study period. Variables assessed include age, sex, household mosquito exposure (continuous in log and categorical 0-8,>8-40, >40-80, >80), and OD values from 27 antibodies: the optimum lambdas were selected using 5-fold cross-validation that achieves the minimum Poisson deviance. The relative variable importance of selected variables was described using standardised coefficients. Only children with complete information were included in the analyses (complete case analyses, n = 211). R statistical software version 4.0.2 with R package 'glmnet' (version 4.1.4) was used[73].

**Reporting summary**. Further information on research design is available in the Nature Research Reporting Summary linked to this article.

## Data availability

All data generated or analysed during this study are included in this published article (and its supplementary information files). Source data are provided as a Source Data file.

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

## Acknowledgements

We thank Australian Red Cross Blood Service for providing malaria naïve samples. We thank all staff of Infectious Disease Research Collaboration, Uganda. We thank all study volunteers and their guardians. This work was supported by the National Health and Medical Research Council of Australia (Early Career Fellowship 1125656, Career Development Award 1141278, Project Grant 1125656 and Ideas Grant 1181932 to M.J.B.; Senior Research Fellowship 1077636 to J.G.B.). Additional support was provided by NIH (U19 A108974 to G.D., and U01 AI150741 to P.J.). The Burnet Institute is supported by the National Health and Medical Research Council for Independent Research Institutes Infrastructure Support Scheme and the Victorian State Government Operational Infrastructure Support.

## Author contributions

J.A.C., J.R.L., G.M., P.J., M.J.B. designed research study; J.A.C., J.R.L., L.de la P., A.S., D.A., N.D. conducted experiments; J.A.C., J.R.L., S.O., G.H., M.J.B. analysed data; B.D.W., P.M.H., J.G.B. provided essential reagents, technical expertise and contributed to assay development; I.S., M.N., F.N., K.M., B.G., P.T., E.A., P.B., J.R., M.K., G.D., M.F., P.J. conducted and supervised the clinical studies and sampling; J.A.C., J.R.L., G.M., P.J., M.J.B. led manuscript preparation with feedback from all authors.

## Competing interests

The authors declare no competing interests.

## Additional information

**Peer review information** *Nature Communications* thanks Ann Moormann and Other anonymous Reviewer(s) to the peer review of this work. Peer review Reports are available.

