## [Peer Review File · Nature Communications]

REVIEWER COMMENTS

Reviewer #1 (Remarks to the Author):

This is an exciting and important study. The large number of patients enrolled and inclusion of the effect of high vs low exposure, status of parasitemia, and having the household mosquito exposure is a brilliant way to test the role of repetitive or continuous exposure. In addition, the use of cutting-edge assays to assess which FcRs are implicated is great. The finding that Th2 Tfh decrease in healthy control kids with age is also so interesting and consistent with aging immunology. The extent of the thoughtful multi-parameter analysis is also impressive. While interpretation is challenging, the authors do a good job of balancing accurate science writing and hypothesis-based conclusions in the discussion.

The finding that Tfh correlate with cytophilic antibodies in *P. falciparum* is really good, and continuing the collective goal of defining the important Tfh types, and protective correlates of the antibody functions. The conundrum is that cytophilic antibodies are generally associated with Th1 cells and IFN- γ , so understanding what CXCR3⁻ CXCR5⁺ and CXCR3⁺ or CCR6⁺ CXCR5⁺ cells can or cannot do in the context of *Plasmodium* infection is a critical question.

The possible inclusion of Th1-like cells in the Th1-Tfh gate might make it harder to see effects due to Th1-like Tfh versus Tfh-like Th1s. In other words, PD-1^{hi}-int CXCR5⁺CXCR3⁺ cells likely include Th1-like cells, with low Bcl6, so non-Tfh (Carpio et al, 2015), and possibly inhibited antibody help functions, (Ryg-Cornejo, Hansen et al; Zander, Butler et al), some testing of the PD-1^{int} or total CD4 (or CD38⁺ or Ki67⁺) CXCR5⁺/⁻ CXCR3 or CCR6⁺ in the correlation studies might reveal connections to symptoms and/or possibly isotypes or FcR binding also with a more clear mechanistic basis.

The allusion that increasing Tfh, or Th2-Tfh may be detrimental to development of symptoms in future infections, is really interesting. Isotypes have been correlated with disease before, as have cytokine balance, e.g. TNF/IL-10. A recent paper also touches upon the balance of Tfh and Th1 effector cells and their effect on pathology. Carpio et al, 2020 suggests that making T cells more Th1-biased (STAT3 KO) protects against parasitemia in a second infection, but has a tendency to increase the severity or prolong parasitemia in the first infection. It is possible that changes in Tfh subsets also represent a change in Th1-like effector cell numbers or phenotype, or CD8 T cells that can cause symptoms. Consider adding in supplemental data indicating the fractions of total CD4 that are CCR6⁺ and CXCR3⁺, as this skewing may be a more proximal explanation for how Th2- and Th17 -Tfh correlate with disease upon infection.

Minor

Antigen Specificity is clearly required for further investigation of Tfh quality in human malaria, but gating on Ki67+ or CD38+ cells in Pf+ patients should indicate Plasmodium specific phenotypes?

Fig 2. A good control for Ki67 and Icos would be expression on total CD4 to compare total T cells or CD4 with Tfh specific proliferation to know of specific effects occur within Tfh subset.

Fig 5, Ki67 would be great to know if Th2 Tfh proliferation in Pf positive patients, is different perhaps in cluster 4? Markers and presence in the blood change on Tfh a lot after acute activation, an analysis of only positive kids might be informative for malaria-specific phenotypes.

Also, further analysis of household mosquito exposure should be done, e.g. are there fewer individuals from the lowest exposure group in cluster 3 vs 4? More in highest in cluster 1?

Missing references-

Several recent studies have had the goal of identifying Tfh subsets, notably using mass spectrometry, are those cited where relevant (Ionnidis et al; ?. The predominance of Th1 markers on both Tfh and GC Tfh was described almost simultaneously in human and mouse in 2015. Note the work of Carpio and Stephens showing that most of the CXCR5+ CD4 T cells are more like Th1 cells than Tfh (and don't depend on Bcl6). Importantly, this Th1 effector-like population may be clouding the statistics using only CXCR5 and CXCR3, as also suggested by the dominance of NK1.1+ and Ly6c+ effector cells with some ability to help B cells in vitro (Wikenheiser, Zander), and critical roles for their IL-10 and IL-21. In mouse, CXCR6 was shown to be a good marker of IFN-g or T-bet expression (Soon et al), though it is not known if this would work in humans, or maybe bcl6 high would identify true GC Tfh in a study like this? The first to show that antibody in infection including malaria is driven by Tfh, using bcl6 deficient mice, specific to T cells, have persistent infection, and this was done first by the Langhorne group, Perez-Mazliah et al.

Fig 7. follow-up study to know odds ratio of infection for each kid is fantastic, particularly the symptoms over the following year in active follow up! The great thing is that Tfh correlate with directionality in OR analysis with antibodies overall, but the subsets are more challenging to interpret. If possible, explore association with (CD38, Ki67 or PD-1int?) CXCR3+ CD4 and symptoms.

Overall suggestion, could CD38 and Ki67 be used to focus on currently activated cells?

Discussion “Kenya populations” should probably be Kenyan people, right

Reviewer #2 (Remarks to the Author):

This is a well-designed study and a well-written manuscript describing changes in the proportion of follicular helper T cell subsets within a malaria-exposed childhood study population in Uganda. This paper describes three significant findings. First, there is an age-associated shift from Th2 to Th1 Tfh cell subsets independent of malaria exposure, which was confirmed by including age-matched controls from Australia (non-malaria exposed children). Second, they describe a hierarchical acquisition of functional malaria-specific antibodies based on parasite stage, putatively associated with antigen dose. Finally, they tested their Tfh and antibody profiles as predictors of infection and disease in the longitudinal component of the study. They found Tfh-17 as being the only subset associated with protection from infection. Whereas their functional antibody profiles of different compositions were associated with risk of infections (ie exposure risk), rather than protection. They also evaluated symptomatic infections versus asymptomatic infections (controlling for age and current infection) and concluded that only MSP2 IgG3, AMA1 IgM and CSP OPA were associated with reduced odds of symptoms. These findings are important to the field of malaria immunology as well as understanding the age-associated development of human Tfh cell subsets, in general. They also highlighted sex-based differences in antibody profiles which warrants further investigation. The main limitation of this study not including a more comprehensive discussion on how the changing kinetics Tfh subset proportions observed confer protection from infection versus protection from disease. The findings related to cytophilic and functional antibodies with protection from symptomatic infections were consistent with previous studies and therefore not a novel finding.

Comments:

1. Given the significant association between Tfh-17 cells and protection from infection, the discussion lacked any contextual interpretation of this finding.
2. Did the authors measure absolute CD4 T cell and Tfh cell counts? It is well known that total lymphocyte counts change with age. How would this impact the analysis and interpretation of your findings? If this data is not available, then it should be mentioned as a limitation of the study.
3. Symptomatic malaria infections are associated with hypergammaglobulinemia. Did the authors measure total immunoglobulin levels across their study populations? How might this impact the analysis, for example using total IgG as the denominator when measuring differences in antibody levels, and interpretation of your findings.

4. Did the authors measure gametocytes in their study populations? In the Discussion, this is mentioned as being associated with antigen-dose dependent development of antibody responses. Is there any correlation between gametocytes in the blood and Pfs230 responses?
5. The negative findings for Tfr cells (supplemental figure 2) does not add to the significance or impact of this paper and could be omitted. However, this is at the authors discretion if this is a novel negative finding.
6. This may be outside the scope of this study, but did the authors measure IL-10 levels within this study population and correlate them with IgG3 to IgG1 ratios?

Reviewer #3 (Remarks to the Author):

This paper describes a detailed investigation of the interplay between socio-demographic factors, Tfh cells and antibodies, focussing on data from children in a malaria-endemic region in Uganda. The statistical analysis is generally appropriate and clearly presented. I have a few suggestions for the authors to consider which I think could improve the clarity of the paper. Specific comments follow.

1. For context, it would be helpful to include a little more epidemiological information on the cohort of Ugandan children who are the main focus of this work. For example, please could the authors include information on how many children were diagnosed with *P. falciparum* infection (of the different classifications) during follow-up, as well as basic demographic data – this could be in the form of another supplementary table.
2. I found the comparison with the malaria naïve cohort a little simplistic. There are likely to be many other differences between the two cohorts, which could influence the Tfh cell patterns and cause differences or similarities. It could also be noted that although the patterns for Th2-Tfh and Th1-Tfh cells looked similar between the Uganda and Australia groups, the pattern for Th17-Tfh cells looked quite different (i.e. an increase during childhood in the Australian group compared to a plateau in the Ugandan group) – the authors could pick up on this point. In addition, from a technical point of view, I'm not sure it makes sense to fit curves on the Australian cohort including all age groups, but then compare them with curves fitted on the Uganda cohort which only contains 0-11 year olds. In other words, in Figure 1I, it might be helpful to “zoom in” on the patterns specifically in the same age group as is covered in the Uganda cohort i.e. up to 11 years – how would figure 1I look if the LOESS curves were fitted based on the children alone?
3. The authors mention that a complete case analysis was done – how did the 211/212 children included in this complete case analysis differ in terms of known characteristics compared to the 262-211 = 51 (~20%) of children who were not included? Related to this point, could the authors comment on the generalisability of their findings in the discussion?

4. In regression models include HME, did the authors consider fitting household mosquito exposure (HME) as a continuous variable or (if sticking with it as a categorical variable) conducting a test for trend? The patterns shown in some tables, e.g. supplementary table S1 suggest that the relationship of HME with e.g. ICOS+ is approximately linear for Th2 and Th17. I think the associations seen for HME would also be worth a comment in the text (e.g. where Figure 2G and Supplementary Table S1 are referenced).
5. In the PCA analysis, how did the authors decide that four clusters was the most appropriate number for interpretation purposes?
6. Regarding the results showing that antibodies to MSP2 and AMA1 were associated with an increased odds of infection in the following year, the authors state that this is consistent with previous studies showing that antibodies to blood stage malaria parasites can accurately measure exposure risk – did they also find that “household mosquito exposure” was associated with increased odds of infection? I cannot see these results reported anywhere, but apologies if I have missed them.

We thank the reviewers for the positive and constructive feedback. We have addressed all comments as outlined below and believe that the manuscript is significantly improved. We hope that our manuscript will now be acceptable for publication.

We would like to highlight additional supplementary data now included of analysis of total CD4, or CXCR5+ and CXCR5- cells. These data strengthen the manuscript by highlighting that the majority of the age and malaria modulations of Tfh cells (CXCR5+PD1+), and the association between Tfh cells and antibodies are specific to this subset (see Supplementary Figures S2, S4, S6). Combined with other modifications outlined below, we believe that these changes strengthen our findings.

REVIEWER COMMENTS

Reviewer #1 (Remarks to the Author):

This is an exciting and important study. The large number of patients enrolled and inclusion of the effect of high vs low exposure, status of parasitemia, and having the household mosquito exposure is a brilliant way to test the role of repetitive or continuous exposure. In addition, the use of cutting-edge assays to assess which FcRs are implicated is great. The finding that Th2 Tfh decrease in healthy control kids with age is also so interesting and consistent with aging immunology. The extent of the thoughtful multi-parameter analysis is also impressive. While interpretation is challenging, the authors do a good job of balancing accurate science writing and hypothesis-based conclusions in the discussion.

*The finding that Tfh correlate with cytophilic antibodies in *P. falciparum* is really good, and continuing the collective goal of defining the important Tfh types, and protective correlates of the antibody functions. The conundrum is that cytophilic antibodies are generally associated with Th1 cells and IFN- γ , so understanding what CXCR3- CXCR5+ and CXCR3+ or CCR6+ CXCR5+ cells can or cannot do in the context of *Plasmodium* infection is a critical question.*

The possible inclusion of Th1-like cells in the Th1-Tfh gate might make it harder to see effects due to Th1-like Tfh versus Tfh-like Th1s. In other words, PD-1^{hi}-int CXCR5+CXCR3+ cells likely include Th1-like cells, with low Bcl6, so non-Tfh (Carpio et al, 2015), and possibly inhibited antibody help functions, (Ryg-Cornejo, Hansen et al; Zander, Butler et al), some testing of the PD-1^{int} or total CD4 (or CD38+ or Ki67+) CXCR5+/- CXCR3 or CCR6+ in the correlation studies might reveal connections to symptoms and/or possibly isotypes or FcR binding also with a more clear mechanistic basis.

Response: We thank the reviewer for the positive and helpful feedback on our work. We agree that understanding the mechanisms by which Th1 and Tfh cells drive antibody development in malaria is a critical area of research. In humans, CXCR5+PD1+ CD4 T cells are the subset that produces the largest amount of the Tfh cytokine IL21, and most strongly resemble germinal center Tfh cells transcriptionally (Locci et al, 2013). Therefore,

these cells are the focus of our study. This explanation is now included in results section lines 120-123.

We agree with the reviewer that CXCR5+PD1+ CD4 T may contain both Th1-like Tfh cells and Tfh-like Th1 cells. However, due to low Bcl6 expression in human blood Tfh cells, differentiating these two cell populations in humans is problematic. We have addressed the reviewer's query in part by including in the revision both total CXCR5+ and CXCR5- populations in our PCA analysis incorporating all Tfh and antibody responses from all individuals (Supplementary Figure S6.) These data show that the CD4+ cellular differences observed between cluster 3 and 4 were largely unique to Tfh cells and not seen in total CXCR5+ or CXCR5- CD4 T cells. There were no differences in subset distribution of CXCR5+ or CXCR5- cells between cluster 3 and 4, and while there was a small increase in ICOS+ CXCR5+ Th1-like cells in cluster 4, there were no other differences in ICOS or Ki67 expression between the two clusters. This finding is consistent with the hypothesis that activated and/or proliferating Tfh, rather than total Th1 cell populations, are most correlated with functional antibody development. Although we would have like to evaluate PD-1^{int} cells, the PD1 antibody used does not have sufficient sensitivity to analyse PD1 high and intermediate cells confidently. We have added this as a limitation in our discussion section (line 566).

The allusion that increasing Tfh, or Th2-Tfh may be detrimental to development of symptoms in future infections, is really interesting. Isotypes have been correlated with disease before, as have cytokine balance, e.g. TNF/IL-10. A recent paper also touches upon the balance of Tfh and Th1 effector cells and their effect on pathology. Carpio et al, 2020 suggests that making T cells more Th1-biased (STAT3 KO) protects against parasitemia in a second infection, but has a tendency to increase the severity or prolong parasitemia in the first infection. It is possible that changes in Tfh subsets also represent a change in Th1-like effector cell numbers or phenotype, or CD8 T cells that can cause symptoms. Consider adding in supplemental data indicating the fractions of total CD4 that are CCR6+ and CXCR3+, as this skewing may be a more proximal explanation for how Th2- and Th17 -Tfh correlate with disease upon infection.

Response: We now include the fractions of total CD4 that are CCR6+ and CXCR3+ (Supplemental Figure S2), which reveal similar age-dependent remodeling in the total CXCR5+ (PD1-/+) population and CXCR5- CD4 T cells. However, Tfh cells have higher overall Th1-like cells, and reduced Th2 like cells compared to total CXCR5+ or CXCR5- CD4 T cells, suggesting that Tfh cell subset redistribution was driven by both global impacts of age on CD4 T cells, as well as Tfh specific factors (**Supplementary Figure S2**). (lines 138-143).

Minor

Antigen Specificity is clearly required for further investigation of Tfh quality in human malaria, but gating on Ki67+ or CD38+ cells in Pf+ patients should indicate Plasmodium specific phenotypes?

Response: ICOS⁺ and Ki67⁺ Tfh cells are detected in both parasite infected and uninfected individuals (Figure 2C/D). Because of this, we are not confident that all ICOS⁺ or Ki67⁺ Tfh cells in parasite infected individuals are *Plasmodium* antigen specific. Future studies are required to apply methods to detect antigen specific Tfh cells, such as Activation Induced Marker assays, to further understand these issues. We have elaborated on this point in the discussion lines 561-569.

Fig 2. A good control for Ki67 and Icos would be expression on total CD4 to compare total T cells or CD4 with Tfh specific proliferation to know of specific effects occur within Tfh subset.

Response: ICOS and Ki67 expression on CD4 T cells, as analysed for Tfh cells, is now presented in Supplementary Figure S4, lines 192-202. Data show that the majority of changes to ICOS and Ki67 on Tfh cells with age, infection, and household mosquito exposure are unique to Tfh cells and not seen on total CD4 T cells.

Fig 5, Ki67 would be great to know if Th2 Tfh proliferation in Pf positive patients, is different perhaps in cluster 4? Markers and presence in the blood change on Tfh a lot after acute activation, an analysis of only positive kids might be informative for malaria-specific phenotypes.

Response: There were no differences in Ki67 expression on Th2-Tfh cells between parasite positive and negative children within cluster 4. This information is now included in lines 361-363 of the results.

Also, further analysis of household mosquito exposure should be done, e.g. are there fewer individuals from the lowest exposure group in cluster 3 vs 4? More in highest in cluster 1?

Response: There was no statistically significant difference in the frequencies of children in each of the exposure categories in each of the clusters identified (Chi-square 15.47, p=0.07). This data is presented in Figure 5D.

Missing references-

Several recent studies have had the goal of identifying Tfh subsets, notably using mass spectrometry, are those cited where relevant (Ionnidis et al; ?. The predominance of Th1 markers on both Tfh and GC Tfh was described almost simultaneously in human and mouse in 2015. Note the work of Carpio and Stephens showing that most of the CXCR5⁺ CD4 T cells are more like Th1 cells than Tfh (and don't depend on Bcl6). Importantly, this Th1 effector-like population may be clouding the statistics using only CXCR5 and CXCR3, as also suggested by the dominance of NK1.1⁺ and Ly6c⁺ effector cells with some ability to help B cells in vitro (Wikenheiser, Zander), and critical roles for their IL-10 and IL-21. In mouse, CXCR6 was shown to be a good marker of IFN- γ or T-bet expression (Soon et al), though it is not known if this would work in humans, or maybe bcl6 high would identify true GC Tfh in a study like this? The first to show that antibody in infection including malaria is driven by Tfh, using bcl6 deficient mice, specific to T cells, have persistent infection, and this was done first by the Langhorne group, Perez-Mazliah et al.

Response: The introduction has been modified to include the important mice studies indicated that show an essential role of Tfh cells in antibody induction in malaria, along with the Th1-like profiles of these cells (lines 74-76, and 82-84). Additionally, the recent Ioannidis et al, JCI Insight paper is referenced in the discussion regarding Tfh cell subsets expanded during infection (lines 551-552). While we agree with the reviewer that the subset differences between Th1 and Tfh cells in malaria infection is an important issue, a detailed discussion of this is outside the scope of this paper. We have previously discussed these interesting overlaps and key knowledge gaps in a recent review, Soon et al, Open Immunology, 2021.

Fig 7. follow-up study to know odds ratio of infection for each kid is fantastic, particularly the symptoms over the following year in active follow up! The great thing is that Tfh correlate with directionality in OR analysis with antibodies overall, but the subsets are more challenging to interpret. If possible, explore association with (CD38, Ki67 or PD-1int?) CXCR3+ CD4 and symptoms.

Response: In this current analysis, we are focused on Tfh cells, analysed based on CXCR5+ and PD1+ expression, as in humans these cells produce the most amount of IL12 and the activation of these cells has been associated with antibody levels in a number of human infections. As discussed above, our flow cytometry analysis does not have sufficient power to differentiate confidently between PD1hi and PD1int cells, so we have not been able to explore the associations between potential non-Tfh cells as suggested. We would like to also refer the reviewer to our previous publication, Boyle et al, Frontiers Immunology, 2017. In that study, we investigated the association between parasite specific Th1 cells (based on IFN γ production following parasite stimulation) and protection within this cohort. This study showed that Th1-CD4 T cell responses were not associated with odds of infection or symptoms when infected in this population.

Overall suggestion, could CD38 and Ki67 be used to focus on currently activated cells?

Response: ICOS+ and Ki67+ Tfh cells are detected in both parasite infected and uninfected individuals (Figure 2C/D). Because of this, we are not confident that all ICOS+ or Ki67+ Tfh cells in parasite infected individuals are *Plasmodium* antigen specific. Future studies are required to apply methods to detect antigen specific Tfh cells, such as Activation Induced Marker assays, to further understand these issues. We have elaborated on this point in the discussion lines 561-566. CD38 was not measured in our studies.

Discussion "Kenya populations" should probably be Kenyan people, right

Response: This has been changed accordingly in the revised manuscript.

Reviewer #2 (Remarks to the Author):

This is a well-designed study and a well-written manuscript describing changes in the proportion of follicular helper T cell subsets within a malaria-exposed childhood study population in Uganda. This paper describes three significant findings. First, there is an age-associated shift from Th2 to Th1 Tfh cell subsets independent of malaria exposure, which was confirmed by including age-matched controls from Australia (non-malaria exposed children). Second, they describe a hierarchical acquisition of functional malaria-specific antibodies based on parasite stage, putatively associated with antigen dose. Finally, they tested their Tfh and antibody profiles as predictors of infection and disease in the longitudinal component of the study. They found Tfh-17 as being the only subset associated with protection from infection. Whereas their functional antibody profiles of different compositions were associated with risk of infections (ie exposure risk), rather than protection. They also evaluated symptomatic infections versus asymptomatic infections (controlling for age and current infection) and concluded that only MSP2 IgG3, AMA1 IgM and CSP OPA were associated with reduced odds of symptoms. These findings are important to the field of malaria immunology as well as understanding the age-associated development of human Tfh cell subsets, in general. They also highlighted sex-based differences in antibody profiles which warrants further investigation. The main limitation of this study not including a more comprehensive discussion on how the changing kinetics Tfh subset proportions observed confer protection from infection versus protection from disease. The findings related to cytophilic and functional antibodies with protection from symptomatic infections were consistent with previous studies and therefore not a novel finding.

Comments:

1. *Given the significant association between Tfh-17 cells and protection from infection, the discussion lacked any contextual interpretation of this finding.*

Response: We agree that the association between Th17-Tfh cells and protection from infection requires further contextualization. We have added the following to the discussion (lines 542-551).

“Levels of blood stage antibodies were also associated with increased odds of infection, this is consistent with previous studies showing that antibodies to blood stage malaria parasites can accurately measure exposure risk ^{42,43}. Interestingly, we also observed that increased proportions of Th17-Tfh cells were associated with a reduced odds of infection, but not symptoms once infected. While it is possible that Th17-Tfh cells may have a role in protecting from blood stage infection measured here, we hypothesize that this association is instead driven by exposure. Children with higher malaria exposure have a proportional decrease in Th17-Tfh cells as malaria specific Th1- and Th2-Tfh subsets expand, consistent with the proliferation of Th1- and Th2- subsets during active infection seen here and reported previously ^{14,15}.”

2. *Did the authors measure absolute CD4 T cell and Tfh cell counts? It is well known that total lymphocyte counts change with age. How would this impact the analysis and interpretation of your findings? If this data is not available, then it should be mentioned as a limitation of the study.*

Response: Absolute counts are not available for this cohort. While it is recognized that CD4 T cell concentrations reduce with age in children, it is unknown whether these changes to cell concentrations are linked to function and/or protections from disease. We have noted this limitation in the discussion, lines 572-575.

3. Symptomatic malaria infections are associated with hypergammaglobulinemia. Did the authors measure total immunoglobulin levels across their study populations? How might this impact the analysis, for example using total IgG as the denominator when measuring differences in antibody levels, and interpretation of your findings.

Response: While total IgG concentrations in serum can vary between people, our primary interest is comparing relative IgG reactivity and functions to different malaria antigens between groups, or correlations between response variables. The approach we have used is valid for those aims. The total concentration of IgG in serum will not have a direct effect on the specific IgG binding to malaria antigens. In previous studies, we have investigated methods to quantify antigen-specific antibody concentrations but we have not identified a reliable method that can be applied in large sample sets or with small sample volumes.

4. Did the authors measure gametocytes in their study populations? In the Discussion, this is mentioned as being associated with antigen-dose dependent development of antibody responses. Is there any correlation between gametocytes in the blood and Pfs230 responses?

Response: The presence of gametocytes was assessed by blood smear. Only one child in the cohort had detected gametocytes at time of sampling. This is now indicated in the results lines 257-259, and described in methods, line 605.

5. The negative findings for Tfr cells (supplemental figure 2) does not add to the significance or impact of this paper and could be omitted. However, this is at the authors discretion if this is a novel negative finding.

Response: The Tfr data in supplementary has remained in the submission. We believe this novel negative finding may be of interest to some readers.

6. This may be outside the scope of this study, but did the authors measure IL-10 levels within this study population and correlate them with IgG3 to IgG1 ratios?

Response: IL10 serum levels have not been measured in this cohort. However, we do not believe that this would be informative on specific IgG1/IgG3 ratios of antibodies to malaria. In malaria responses, IgG1/IgG3 ratios are largely dependent on the specific antigen target. For example, here we measure antibodies to MSP2 and AMA1 which have strikingly different IgG1/IgG3 ratios, with MSP2 antibodies IgG3 dominant, and AMA1 antibodies IgG1 dominant (Figure 3). This suggests that the cytokine milieu that drives IgG1/IgG3 ratios is highly localized and specific to individually targeted B cells.

Reviewer #3 (Remarks to the Author):

This paper describes a detailed investigation of the interplay between socio-demographic factors, Tfh cells and antibodies, focussing on data from children in a malaria-endemic region in Uganda. The statistical analysis is generally appropriate and clearly presented. I have a few suggestions for the authors to consider which I think could improve the clarity of the paper.

Specific comments follow.

*1. For context, it would be helpful to include a little more epidemiological information on the cohort of Ugandan children who are the main focus of this work. For example, please could the authors include information on how many children were diagnosed with *P. falciparum* infection (of the different classifications) during follow-up, as well as basic demographic data – this could be in the form of another supplementary table.*

Response: We have now included participant demographics and malaria information in Supplementary Table S1.

2. I found the comparison with the malaria naïve cohort a little simplistic. There are likely to be many other differences between the two cohorts, which could influence the Tfh cell patterns and cause differences or similarities. It could also be noted that although the patterns for Th2-Tfh and Th1-Tfh cells looked similar between the Uganda and Australia groups, the pattern for Th17-Tfh cells looked quite different (i.e. an increase during childhood in the Australian group compared to a plateau in the Ugandan group) – the authors could pick up on this point. In addition, from a technical point of view, I'm not sure it makes sense to fit curves on the Australian cohort including all age groups, but then compare them with curves fitted on the Uganda cohort which only contains 0-11 year olds. In other words, in Figure 11, it might be helpful to “zoom in” on the patterns specifically in the same age group as is covered in the Uganda cohort i.e. up to 11 years – how would figure 11 look if the LOESS curves were fitted based on the children alone?

Response: We agree that there are multiple differences between the two cohorts used in our study. However, the purpose of the presented analysis was to investigate whether the decline in Th2-Tfh cells seen in Ugandan children was potentially driven by age, or could be attributed to malaria infection which is co-linear with age in this population. Consistent with a role of age, Th2-Tfh cells declined with age in the malaria naive population. We now modified the text to include the R/p values for the association between age and Th2-Tfh cells in ‘zoomed’ in analysis of children and adults (age 0-15 years and 15+ years respectively). This supports a decline of Th2-Tfh cells in children - $R = -0.74$, $p = 0.01$ in children aged 0 to 15 and $R = -0.19$, $p = 0.6$ in adults age 15 years or greater (lines 159-161).

3. The authors mention that a complete case analysis was done – how did the 211/212 children included in this complete case analysis differ in terms of known characteristics compared to the

262-211 = 51 (~20%) of children who were not included? Related to this point, could the authors comment on the generalisability of their findings in the discussion?

Response: Demographics and malaria exposure characteristics of both the full cohort test for antibodies (n=262) and complete cohort tested for antibodies and Tfh cell responses (n=212) is now presented in Supplementary Table S1.

Regarding generalisability of our findings, we have now included in discussion on limitations that our cohort is in an area of very high malaria burden, and further studies in low and moderate transmission settings are needed (lines 558-560).

4. In regression models include HME, did the authors consider fitting household mosquito exposure (HME) as a continuous variable or (if sticking with it as a categorical variable) conducting a test for trend? The patterns shown in some tables, e.g. supplementary table S1 suggest that the relationship of HME with e.g. ICOS+ is approximately linear for Th2 and Th17. I think the associations seen for HME would also be worth a comment in the text (e.g. where Figure 2G and Supplementary Table S1 are referenced).

Response: The categorical variable for HME was previously defined in this cohort as it was strongly associated with the odds of infection, and the relationship between HME and infection odds is non-linear (Boyle et al, Frontiers Immunology, 2017 - HME was previously described as daily mosquito exposure rate). This information is now included in the methods section, lines XX. For parsimony, we kept this definition in the present analysis. As suggested, testing for trend indicated that ICOS expression was associated with HME in Th1-Tfh cells, but not other subsets. We have now included the following comment in results section lines 211-217.

“Highest household mosquito exposure (<80) was associated with increased ICOS and Ki67 expression on Th1-Tfh cells in univariate analysis (**Figure 2G, Supplementary Table S2**). While this association wasn't maintained once controlling for age and current infection, increasing household mosquito exposure was associated with increased ICOS expression on Th1-Tfh cells when included as a linearised variable, suggesting that higher exposure may contribute to activation of Th1 cells (Coef 1.87, p=0.04).”

5. In the PCA analysis, how did the authors decide that four clusters was the most appropriate number for interpretation purposes?

Response: Four clusters were chosen based on Hartigan's rule with an inclusion value of ~10, and sample size consideration. This information is now included in the methods lines 730-731.

6. Regarding the results showing that antibodies to MSP2 and AMA1 were associated with an increased odds of infection in the following year, the authors state that this is consistent with previous studies showing that antibodies to blood stage malaria parasites can accurately measure

exposure risk – did they also find that “household mosquito exposure” was associated with increased odds of infection? I cannot see these results reported anywhere, but apologies if I have missed them.

Response: We apologize for this admission. The association between HME and infection risk in this cohort of children has previously been reported in Boyle et al, *Frontiers Immunology*, 2017. This information is now included in results, lines 412 and methods lines 758 and 766.

REVIEWERS' COMMENTS

Reviewer #1 (Remarks to the Author):

All concerns have been addressed

Reviewer #2 (Remarks to the Author):

The authors have adequately address all the reviewers comments and revised their manuscript accordingly.

Ann Moormann

Reviewer #3 (Remarks to the Author):

The authors have done a great job of responding to my comments (and I believe also those of other reviewers). I have no further suggestions or comments. REVIEWERS' COMMENTS

Reviewer #1 (Remarks to the Author):

All concerns have been addressed

Reviewer #2 (Remarks to the Author):

The authors have adequately address all the reviewers comments and revised their manuscript accordingly.

Ann Moormann

Reviewer #3 (Remarks to the Author):

The authors have done a great job of responding to my comments (and I believe also those of other reviewers). I have no further suggestions or comments.

REVIEWERS' COMMENTS

We thank the reviewers for helpful feedback, and note that no further issues were raised.

Reviewer #1 (Remarks to the Author):

All concerns have been addressed

Reviewer #2 (Remarks to the Author):

The authors have adequately address all the reviewers comments and revised their manuscript accordingly.

Ann Moormann

Reviewer #3 (Remarks to the Author):

The authors have done a great job of responding to my comments (and I believe also those of other reviewers). I have no further suggestions or comments.